# HyDesign: a tool for sizing optimization of grid-connected hybrid power plants including wind, solar photovoltaic, and Li-ion batteries

Juan Pablo Murcia Leon[1], Hajar Habbou[1], Mikkel Friis-Møller[1], Megha Gupta[1], Rujie Zhu[1], and Kaushik Das[1]

[1]Department of Wind and Energy Systems, Technical University of Denmark, 4000 Roskilde, Denmark

**Correspondence:** Juan Pablo Murcia (jumu@dtu.dk)

**Abstract.** Hybrid renewable power plants consisting of collocated wind, solar photo-voltaic (PV), and Lithium-ion battery storage connected behind a single grid connection can provide additional value to the owners and society in comparison to individual technology plants such as only wind or only PV. The hybrid power plants considered in this article are connected to the grid and share electrical infrastructure costs across the different generation and storing technologies. In this article, we

propose a methodology for sizing hybrid power plants as a nested optimization problem: with an outer sizing optimization and an internal operation optimization. The outer sizing optimization maximizes the net present values over capital expenditures and compares it with standard designs that minimize the levelized cost of energy. The sizing problem formulation includes turbine selection (in terms of rated power, specific power, and hub height), a wind plant wake losses surrogate, simplified wind and PV degradation models, battery degradation, and an internal energy management system operation optimization. The outer

sizing optimization problem is solved using a new parallel "efficient global optimization" algorithm. This new algorithm is a surrogate-based optimization method that ensures a minimal number of model evaluations but ensures a global scope in the optimization. The methodology presented in this article is available in an open-source tool called HyDesign. The hybrid sizing algorithm is applied for a peak-power plant use case at different locations in India where renewable energy auctions impose a monetary penalty when energy is not supplied at peak hours. We compare the hybrid power plant sizing results when using

two different objective functions: the levelized cost of energy ($LCoE$) or the relative net present value with respect to the total capital expenditure costs ($NPV/C_H$). Battery storage is installed only on $NPV/C_H$-based designs, while hybrid, including wind, solar, and battery, only occurs on the site with good wind resources. Wind turbine selection on this site prioritizes cheaper turbines with lower hub height and lower rated power. The number of batteries replaced changes on the different sites, ranging between two or three units over the lifetime. A significant over-sizing of the generation in comparison to the grid connection

occurs on all $NPV/C_H$-based designs. As expected $LCoE$-based designs are single technology with no batteries.

## 1  Introduction

A hybrid power plant (HPP) consisting of collocated wind, PV, and Lithium-ion battery storage connected behind a single grid connection point can provide better returns of investment than individual source (wind or solar) plants in locations where the wind and solar resources are comparable and for electricity markets in which fixed power purchase agreement electricity prices

are not possible. HPP can be designed to have operational flexibility in terms of dispatchability and ancillary service provision that makes them closer to traditional power plants in terms of achieving additional profitability in markets with time-varying electricity prices under grid connection constraints and that have reduced costs due to the shared infrastructure (Gorman et al., 2020; Dykes et al., 2020).

Sizing of HPP is a Multi-disciplinary Design Analysis and Optimization (MDAO) problem that requires detailed modeling
of the wind and solar resources as well as the wind, PV, and storage performance, costs, and operation (Dykes et al., 2020). Additionally, the selection of the wind turbine (WT) characteristics (specific power, hub height) and PV characteristics (panel orientation) are additional degrees of freedom that can significantly modify the results of the sizing. Traditional objective functions of the sizing optimization problem are maximizing net annual energy production or minimizing $LCoE$ (Tripp et al., 2022), but, in general, HPP designs that include energy storage can produce more revenues relative to the cost increase. In this
article, we compare HPP sizing optimization for both $LCoE$ and relative net revenues as objective functions.

A detailed energy management system (EMS) is required to determine the operation of the battery given the time series of wind and solar generation and the battery's capacity. EMS optimization will determine when to charge and discharge the battery with the objective of maximizing the revenue obtained by the HPP. Several articles focus on formulating EMS optimization problems and propose different formulations Al-Lawati et al. (2021); Das et al. (2020); Khaloie et al. (2021a, b); Wang et al.
(2019). Different levels of complexity can be studied in the implementation of EMS, such as (1) rule-based algorithms that prescribe the operation of the battery, (2) deterministic EMS optimization that maximizes the revenues assuming perfect forecasts (full future knowledge) on the price of electricity, the wind and solar generation time-series, (3) robust optimization of EMS operation will provide battery operation under worst case scenarios of forecast errors of generation and prices time-series, and (4) Stochastic optimization of EMS operation that will provide best operation over the entire distribution of forecasting error.
EMS operational optimization within the HPP sizing optimization is not common in the literature, but it is required to unravel the value of HPP fully.

Furthermore, HPP sizing requires solving the long-term performance of the different components through the lifetime of the HPP; this implies modeling the degradation in the performance of the individual components. Li-ion batteries, wind turbines, and PV cells have significant degradation over time. Several models of PV degradation exist (Jordan et al., 2016), and PV
manufacturers can provide a warranty of the degradation curve, while recent publications report measured PV degradation rates (Theristis et al., 2023, 2020). Wind turbine degradation is significantly more complex than performance degradation, e.x. due to blade erosion (López et al., 2023; Panthi and Iungo, 2023; Bech et al., 2018), is compensated by the internal wind turbine pitch control system. Several studies report different levels of wind plant degradation as losses of capacity factor over age (Hamilton et al., 2020; Jia et al., 2016; Staffell and Green, 2014; Astolfi et al., 2022).
Typically, battery cells have to be replaced when their capacity degrades beyond a manufacturer-defined safety threshold. The higher costs due to battery replacement play a dominant role in battery total costs. Therefore, considering battery degradation when sizing HPP can optimize the use of batteries, extending battery lifetime and reducing costs. Battery degradation is a complicated chemical process. Theoretical studies (Safari et al., 2008; Vetter et al., 2005) on battery degradation explain the detailed degradation mechanism of battery cells. However, the required parameters and conditions of the battery cell can not

be obtained in the sizing stage. To incorporate the battery degradation model into the sizing problem, it is possible to use semi-empirical models (Xu et al., 2016) that only require the state of charge time-series (SoC) as input to assess battery lifetime. This model considers the solid electrolyte inter-phase film formation theory calibrated based on experimental observations and it can describe the non-linear degradation process.

To the authors' knowledge, there is no available sizing methodology for the design of utility-scale grid-constrained hybrid

power plants considering all the above-mentioned characteristics. This article presents a general methodology for hybrid plant sizing as a nested optimization, including several novel aspects: (1) turbine selection, (2) PV and wind degradation, (3) internal EMS operation optimization, and (4) battery degradation based on resulting load-cycles. We apply the methodology and report the detailed result of the hybrid plant design in three different locations in India: (a) solar dominant site (b) wind dominant site and (c) low wind and solar resources. The research objective is to build a framework for optimization of hybrid power plants

that is flexible, modular and that can be extended to solve sizing and physical design of HPP.

India is a large market in which HPPs could become important because of the need to provide renewable energy that supports the demand patterns and because of the intermediate solar and wind resources. For this reason, Indian sites are used as example cases in this article.

## 2  Methodology

The design of a HPP is an optimization problem that involves several sub-optimization problems such as: WT selection, wind power plant (WPP) siting and layout optimization, PV array sitting, EMS operation optimization coupled with battery degradation, and electrical infrastructure optimization. HPP sizing optimization focused on maximizing the viability of a HPP installation in a given location requires a simplified approach. The XDSM diagram of the proposed nested optimization for HPP sizing is presented in Fig. 1. In this sizing optimization formulation several simplifications have been performed to reduce

the complexity of the optimization: (1) The WT Layout optimization is replaced by a surrogate of the wakes of sub-optimal WPP. (2) Uncoupled battery, wind, and PV degradation models are used to reduce the complexity of the EMS optimization: the internal operation optimization solves a short-term EMS problem without considering battery degradation but with a penalty for battery power ramping; while a long-term operation rule-based EMS (EMS Long-term) corrects the ideal battery operation for degradation and forecast errors. (3) Simplified electrical infrastructure costs are used, instead of an electrical cable and

infrastructure optimization. (4) No interaction between WT and PV is assumed, neglecting PV losses due to shadow and flickering and changes in the wind boundary layer due to the presence of large PV arrays.

### 2.1  HPP sizing optimization

The HPP sizing optimization problem consists of minimizing $LCoE$ or maximizing $NPV/C_H$ by changing the design variables: rotor-tip to ground height clearance ($h_c$ in meters), turbine's specific power ($sp$ in megawatt per squared meter), turbine's

rated power ($P_{\text{rated}}$ in megawatt), number of wind turbines ($N_{\text{WT}}$), wind's installation density ($\rho_W$ in megawatt per squared kilometer), solar capacity ($S_{\text{MW}}$ in megawatt), PV tilt angle ($\theta_{\text{tilt}}$ in degrees), PV azimuth angle ($\theta_{\text{azim}}$ in degrees), PV inverter

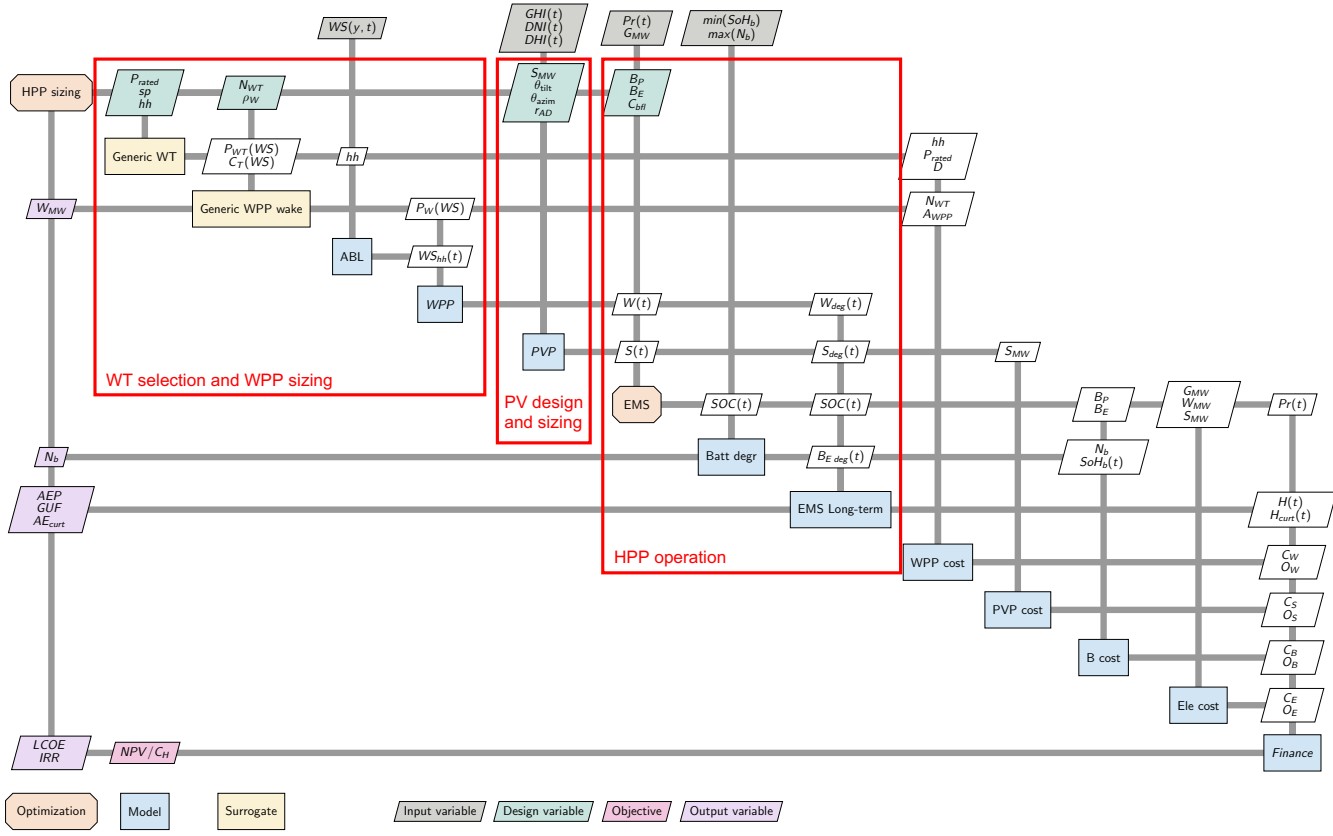

**Figure 1.** HPP sizing as a nested optimization. XDSM diagram.

AC to DC ratio ($r_{AD}$), battery power capacity ($B_P$ in megawatt), battery energy storage capacity in hours at battery power capacity ($B_{Eh}$) and battery fluctuation penalty factor ($C_{bfl}$). Furthermore, the sizing can be forced to only take integer values on some specific design variables such as $N_{WT}$.

$$\min \quad y(x) = \begin{cases} -NPV/C_H(x) \\ LCoE(x) \end{cases} \tag{1}$$

$$x = [h_c, sp, P_{\text{rated}}, N_{\text{WT}}, \rho_W, S_{\text{MW}}, \theta_{\text{tilt}}, \theta_{\text{azim}}, r_{AD}, B_P, B_{Eh}, C_{bfl}]$$

### 2.2 Generic Wind Turbine

A look-up table is built based on DTU's PyWake generic turbine model (Pedersen et al., 2023). The interpolation of this data is a surrogate that predicts the power and thrust coefficient curves given the turbine's specific power, defined as the ratio between the rated power and the rotor area ($sp = P_{\text{rated}}/A$). The wind turbine power curve and thrust coefficient curves are represented

as $P_{WT}(\text{WS})$ and $C_T(\text{WS})$ in Fig. 1. Examples of the surrogate power and thrust coefficient curves are given in Fig. 2. The rotor diameter ($D = 2\sqrt{P_{\text{rated}}/(\pi\, sp)}$) and hub height ($hh = h_c + D/2$) can be computed based on $sp$ and the clearance height.

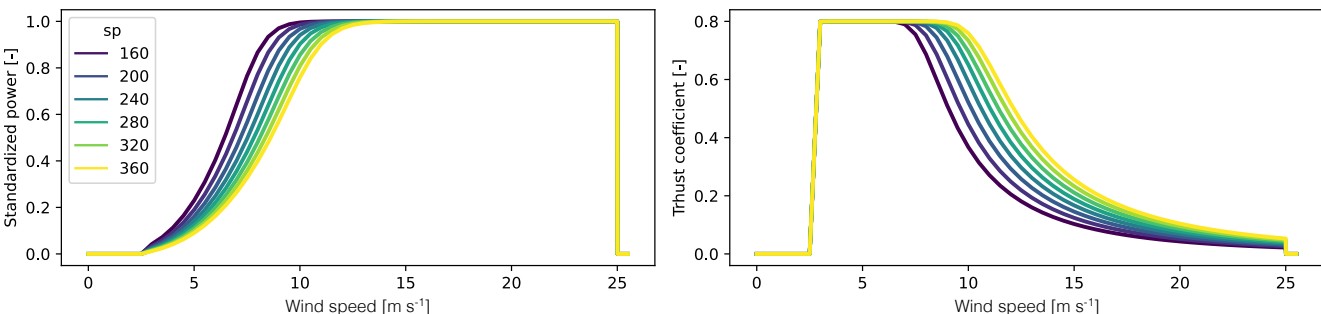

**Figure 2.** Generic wind turbine surrogate: (left) Power curve (right) Thrust curve.

### 2.3 Generic Wind Power Plant Wake Model

A database of wind power plants is generated using circular plant borders and a simplified layout optimization that maximizes the distance between the turbines. Two example layouts are presented in Fig. 3. Here it can be seen that the layouts are sym-
105 metric, and the minimum WT spacing is the consequence of specifying the number of turbines ($N_{\text{WT}}$), the turbine rated power ($P_{\text{rated}}$) and the installation density ($\rho_W$, plant-rated power over the land use area, [MW/km$^2$]). Wakes are simulated using *Py-Wake*'s implementation of Zong's wake model (Pedersen et al., 2023; Zong and Porté-Agel, 2020) which combines a Gaussian wind speed deficit with local turbulence dependent linear wake expansion, with squared sum wake deficit superposition model and Frandsen's added turbulence model as specified in the IEC wind turbine design standard (IEC, 2017).
Detailed wake losses as a function of wind speed and wind direction are simulated for multiple WPP layouts with the same number of turbines ($N_{\text{WT}}$) and installation density ($\rho_W$) for a given WT's specific power, hence given power and thrust curves. The resulting wake losses are aggregated taking the 90-th larger quantile across wind directions and across 20 layouts generated using a different random seed number. A surrogate of the wake losses curve as a function of the hub height wind speed ($\text{WL}(\text{WS})$) is built as a function of the installation density, number of turbines, and specific power of the turbine. Example
results of the surrogate are presented in Fig. 4. Finally, the generic wind plant model will combine the turbine power curve with the expected wake losses to provide a wake-affected plant power curve, see Eq. (2).

$$
\begin{aligned}
W_{\text{MW}} &= N_{\text{WT}}\, P_{\text{rated}} \\
\text{WL}(\text{WS}) &\approx \hat{\text{WL}}(N_{\text{WT}}, sp, \rho_W, \text{WS}) \\
P_W(\text{WS}) &= N_{\text{WT}} \times P_{WT}(\text{WS}) \times (1 - \text{WL}(\text{WS}))
\end{aligned}
\tag{2}
$$

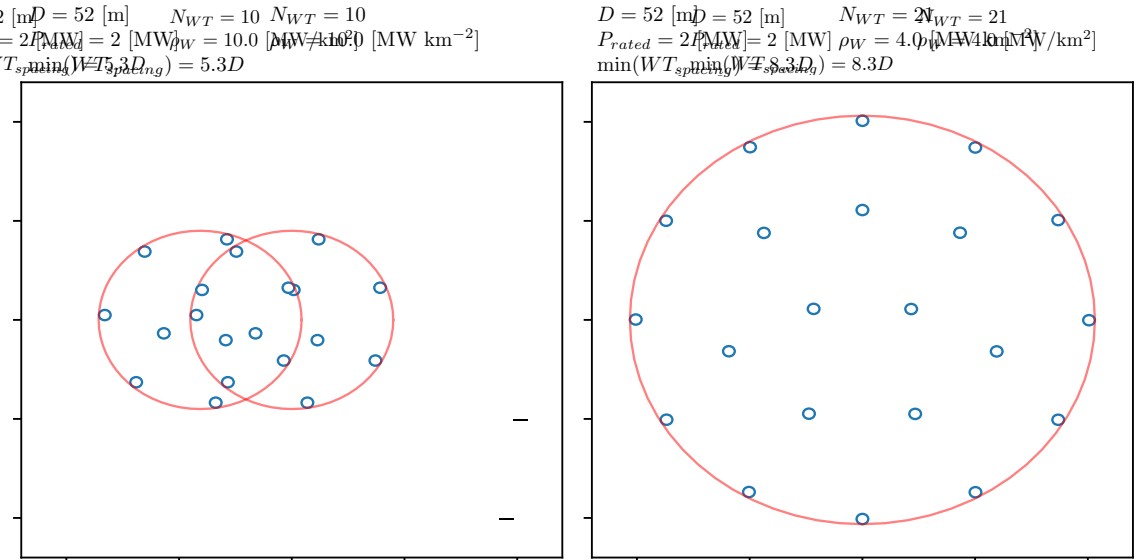

$D = 52$ [m] $N_{WT} = 10$
$P_{rated} = 2$ [MW] $\rho_W = 10.0$ [MW km$^{-2}$]
$\min(WT_{spacing}) = 5.3D$

$D = 52$ [m] $N_{WT} = 21$
$P_{rated} = 2$ [MW] $\rho_W = 4.0$ [MW/km$^2$]
$\min(WT_{spacing}) = 8.3D$

**Figure 3.** WPP example of generated layouts.

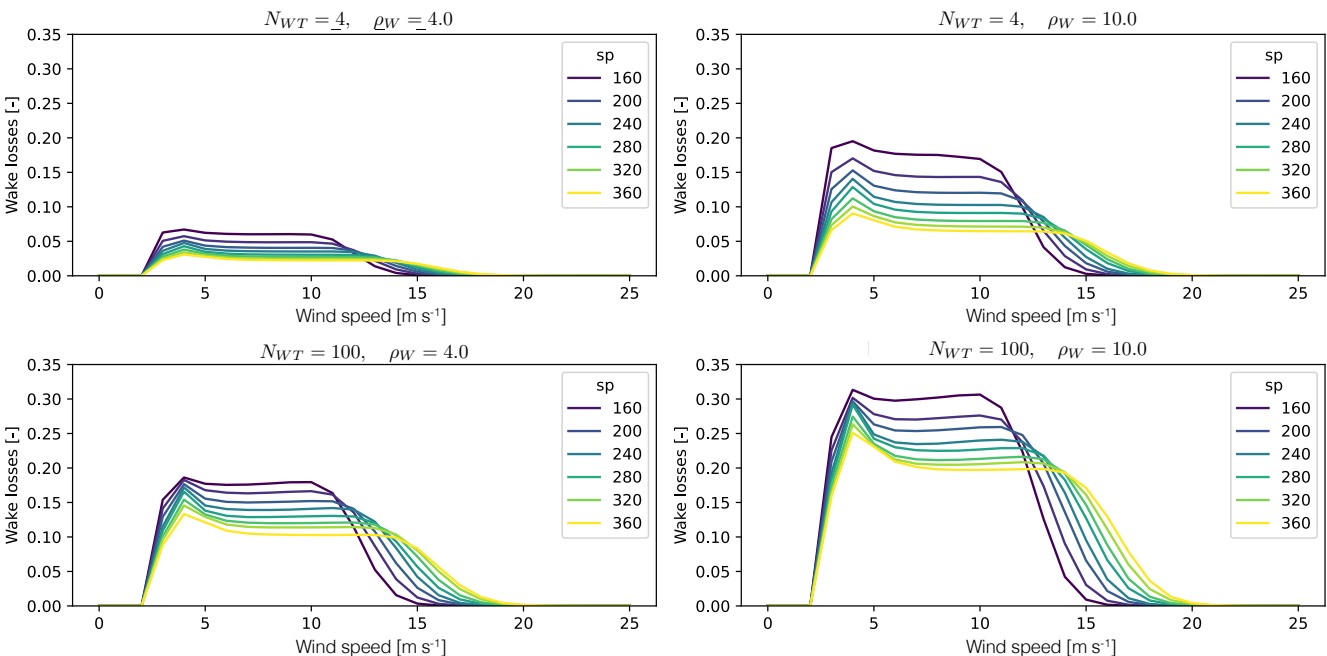

**Figure 4.** Example wake losses as a function of the number of turbines, installation density and WT's specific power.

## 2.4 Weather

ERA5 (Hersbach et al., 2020) is used as a reanalysis dataset for wind resource calculations. The hourly wind velocity time series with a 0.25x0.25 degree resolution in latitude and longitude are interpolated into heights of 50, 100, 150, and 200 [m]. This dataset is stored and interpolated at the location of hybrid power plants using linear interpolation in the horizontal coordinates, keeping the hub height dimension of the velocities to compute the effect of changing the hub height of the turbines in the optimization.

The mean wind speed from the Global Wind Atlas 2 (GWA2) is used for correcting ERA5's mean wind speed following the approach presented in (Murcia et al., 2022). This scaling correction is necessary to include the first-order effects of terrain. The corrected wind speed time series is provided on multiple heights ($\text{WS}(y,t)$) to the atmospheric boundary layer (ABL) model. This model uses a piece-wise power law interpolation to determine the wind speed time series at hub height ($\text{WS}_{hh}(t)$).

ERA5-land is used as a reanalysis of the hourly global horizontal irradiance time-series ($\text{GHI}(t)$) because it has a higher horizontal resolution than ERA5 (0.1x0.1 degree), and it shows better validation metrics for individual PV plant generation modeling (Camargo and Schmidt, 2020). Decomposition of GHI to direct normal irradiance (DNI) and diffuse horizontal irradiance (DHI) is done in two steps: the DISC model is used to estimate the DNI (Maxwell, 1987) using the GHI and relative air mass model based (Kasten and Young, 1989). While the DHI is estimated using the solar position ($\theta_{\text{zenith}}(t)$), see Eq. (3).

$$\text{DHI}(t) = \text{GHI}(t) - \text{DNI}(t) \times \cos(\theta_{\text{zenith}}(t)) \tag{3}$$

## 2.5 Wind power plant model (WPP)

The wind generation time series ($W(t)$) is obtained by interpolating the plant power curve at the hub height's wind speed time series, scaling the generation by the installed capacity. Additionally, efficiency is assumed to cover the electrical and availability losses, see Eq. (4).

$$W(t) = N_{\text{WT}} \times P_{\text{rated}} \times P_W(\text{WS}_{hh}(t)) \times \eta_W \tag{4}$$

Wind turbine degradation is modeled as a mixture of two performance degradation mechanisms: (a) a shift in the power curve towards higher wind speeds represents blade degradation and increasing friction losses (López et al., 2023). (b) a loss factor applied to the power time series represents an increase in availability losses. These mechanisms are depicted on the top plots in Fig. 5. The WT degradation curve ($dl_W(t)$) prescribes the level of loss in capacity factor over time, and the power generation with degradation ($W_{\text{deg}}(t)$) is obtained by linear interpolation of the generation time-series of the new ($W_{\text{new}}$) and fully degraded ($W_{\text{fg}}$) generations, see Eq. (5). A linear degradation on the wind turbine has been used in the study cases; see Fig. 5.

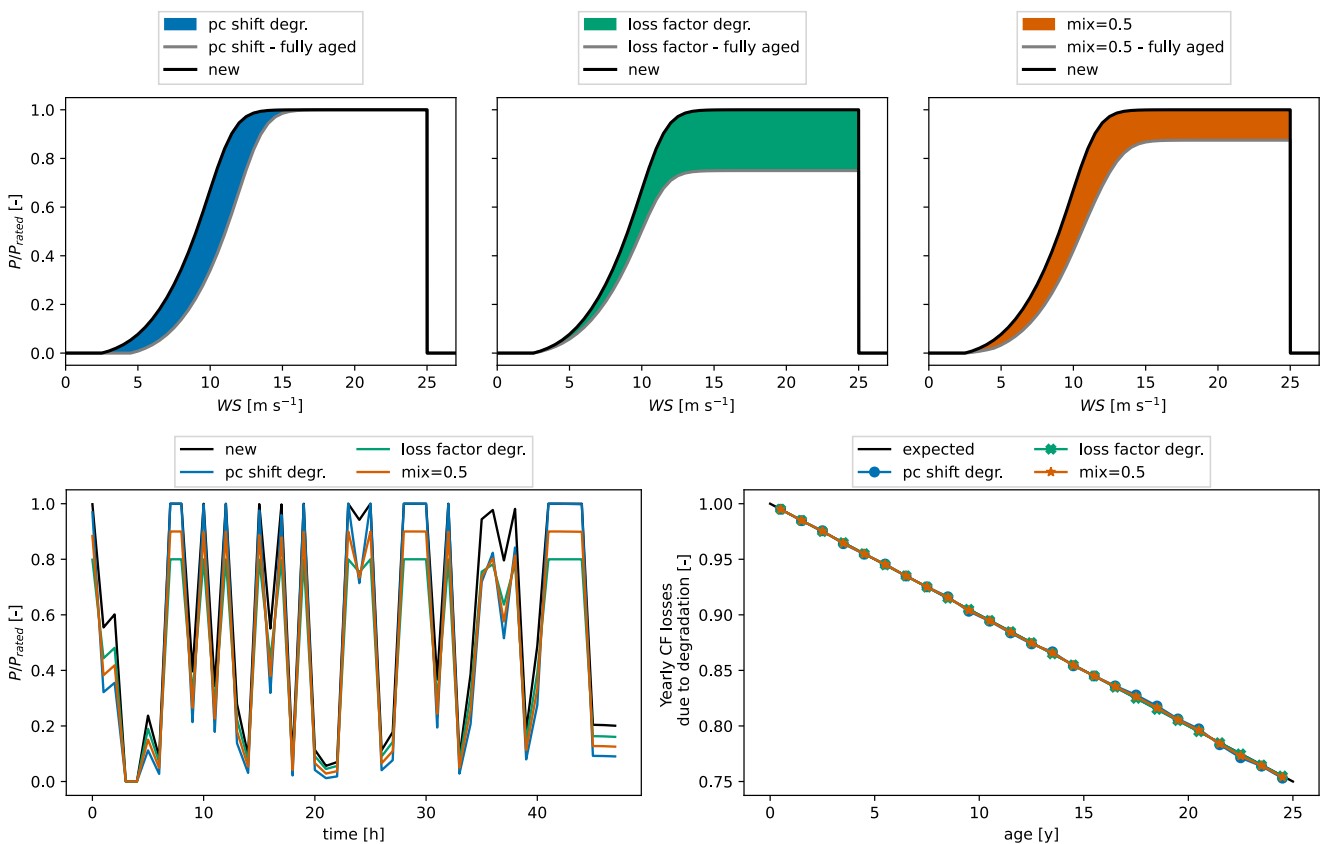

**Figure 5.** (top) Mechanisms of WT degradation (left) shift in power curve (center) loss factor (right) 50%-50% mixture of both mechanisms. (bottom left) Example of 2 days of WPP generation time-series after 20 years. (bottom right) Prescribed degradation curve and resulting losses in CF over the WPP lifetime with the three mechanisms of WT degradation.

$$\alpha(t) = dl_W(t) / \max(dl_W(t))$$
$$W_{\text{deg}}(t) = (1 - \alpha(t)) \times W_{\text{new}}(t) + \alpha(t) \times W_{\text{fg}}(t)$$

(5)

## 2.6 PV power plant model (PVP)

Power conversion uses PVLib (Holmgren et al., 2018) based on a generic 1MW PV plant configuration (PV module, inverter, and open rack with glass-glass) with the irradiance projection transposition model (Davies and Hay, 1978), the Sandia array performance model (SAPM) (King et al., 2004), and the Sandia performance model for grid-connected PV-inverter model (King et al., 2007). The final PV generation requires the PV plant capacity ($S_{\text{MW}}$), the orientation of the panels in terms of tilt and azimuth angles ($\theta_{\text{tilt}}, \theta_{\text{azim}}$), the ratio between DC and AC sides of the inverter ($r_{DA}$), the irradiances (DNI, DHI), the wind speed close to ground (WS$_1(t)$) and the ambient temperature ($T_1(t)$), see Eq. (6).

$$S(t) = S_{\text{MW}} \times \text{PV}(\theta_{\text{tilt}}, \theta_{\text{azim}}, r_{\text{AD}}, \text{DNI}(t), \text{DHI}(t), \text{WS}_1(t), T_1(t)) \tag{6}$$

The PV degradation model is a loss factor that follows a prescribed PV degradation curve $dl_S(t)$. The solar generation time series with degradation is obtained by applying the loss factor to the generation, see Eq. (7). A linear degradation curve is used in the study cases.

$$S_{\text{deg}}(t) = dl_S(t)S(t) \tag{7}$$

## 2.7 Electricity price

The electricity price time series in the spot market ($Pr(t)$) is an input to the model; note that the price time series needs to be correlated with the weather time series. This article focuses on the valuation of time-varying power purchase agreements as the ones that have been seen in the Indian HPP market. This price signal has two levels of electricity price at peak and non-peak (high demand) hours. An example of the peak non-peak PPA electricity price is presented in Fig. 6.

## 2.8 Energy management system optimization model (EMS)

The energy management system optimization model determines the optimal amount of battery charge/discharge and power curtailment that maximizes the revenue generated by the plant over a period of time, including a possible penalty for not meeting the requirement of energy generation and a penalty for battery power ramping to control the number of battery load cycles, see Eq. (8). The EMS optimization is solved using linear programming, applying a piece-wise linearization for the change of battery efficiency in charge and discharge and to the absolute value of the battery power fluctuations. The EMS optimization does not account for battery, WT, or PV degradation, and uses the generations without degradation. Furthermore, the EMS operation optimization assumes perfect knowledge of both the weather and price, and therefore, there are neither forecasting errors on the prices nor the weather.

The revenue is given by the product of electricity price ($Pr(t)$) and the HPP power generation ($H(t)$) minus the penalty over the period ($l$) and minus the battery ramping penalty ($l_b$). The HPP generation is defined as the total power from wind ($W(t)$), PV ($S(t)$), battery charge or discharge ($B(t)$), and power curtailment ($P_{\text{curt}}(t)$).

The penalty ($l$) is the missing energy generated at peak times with respect to the energy requirement over the period ($E_l$) times a mean peak electricity price ($\overline{Pr(t_{\text{peak}})}$). The penalty can only be positive, which means that it can only subtract revenue and generation above the requirement does not yield additional revenue.

The battery fluctuations penalty ($l_b$) is defined as the sum of the products of the absolute battery power fluctuations ($|\Delta B(t)|$) and the difference between peak electricity price and the current price ($Pr_{\text{peak}} - Pr(t)$). This means that large fluctuations in the battery charge/discharge are allowed when the price is high. The battery fluctuation penalty factor ($C_{bfl}$) is a design variable that captures how strongly the battery can be ramped, and therefore, it controls the battery degradation. When $C_{bfl}$ is 0, then large changes in charge/discharge occur, see Fig. 6.

The constraints in the optimization force a minimum level of energy in the battery ($E_{SoC}(t)$) when discharging ($B_{E\,\text{depth}}$), ensure the limits due to batteries power capacity ($B_P$) and energy capacity ($B_E = B_{E\,h}\,B_P$), force the grid capacity ($G$), and include an asymmetric charging/discharging efficiency ($\eta_{\text{charge}}, \eta_{\text{discharge}}$).

$$\max \quad \sum_t \left( Pr(t) \times H(t) \right) - l - l_b$$

$$\text{with} \quad l = \begin{cases} E_l \times \overline{Pr(t_{\text{peak}})} & \text{if} \quad E_l > 0 \\ 0 & \text{if} \quad E_l \leq 0 \end{cases}$$

$$E_l = E_{\text{peak\,req}} - \sum_{t \in t_{\text{peak}}} (H(t)\,\Delta t)$$

$$l_b = C_{bfl} \times \sum_t \left( |\Delta B(t)| \times (Pr_{\text{peak}} - Pr(t)) \right)$$

$$\text{such that } \forall t \quad H(t) = W(t) + S(t) + B(t) - P_{\text{curt}}(t) \tag{8}$$

$$H(t) \leq G$$

$$E_{SoC}(t+1) = \begin{cases} E_{SoC}(t) - \eta_{\text{charge}} B(t)\,\Delta t & \text{if} \quad B(t) \leq 0 \\ E_{SoC}(t) - B(t)\,\Delta t / \eta_{\text{discharge}} & \text{if} \quad B(t) > 0 \end{cases}$$

$$E_{SoC}(t) \geq B_E \times (1 - B_{E\,\text{depth}})$$

$$E_{SoC}(t) \leq B_E$$

$$B(t) \leq B_P$$

$$B(t) \geq -B_P$$

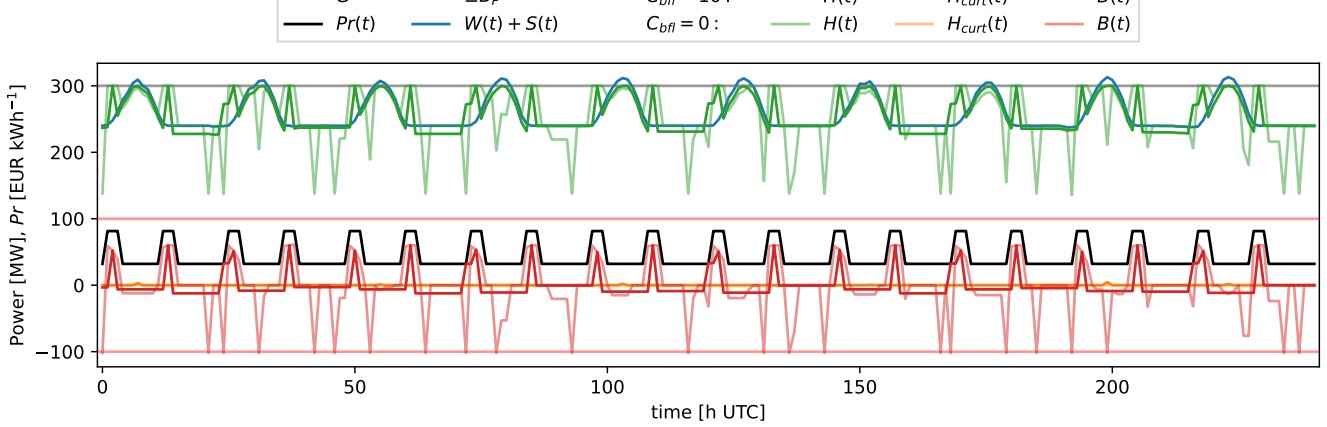

**Figure 6.** EMS comparison in an example HPP for two different battery fluctuation penalty factor $C_{bfl}$.

## 2.9 Battery degradation model

The battery degradation model includes a linear degradation rate as a function of load-cycles and a non-linear degradation due
to the solid electrolyte interphase (SEI) film formation process in the early stage of the battery life. The Rainflow counting
algorithm (Downing and Socie, 1982; Shi et al., 2018) is used to obtain the depth of discharge ($R_{DoD,j}$), mean state of charge
cycle ($R_{SoC,j}$), half or full cycle count ($R_{count,j}$), for a number of load cycles ($j = 1, ..., n_R$) given a relative state of charge
time-series ($E_{SoC}(t)/B_E$). The current age of the battery at each load cycle is defined as $t_{c,j}$.

The linear degradation rate ($f^d$) in Eq. (9) depends on a stress model due to the depth of discharge ($S_{DoD}$), a stress model
due to the age of the battery ($S_t$), a stress model due to the state of charge ($S_{SoC}$), and a stress model due to cell temperature in
Kelvin ($S_T$). The stress factor models are empirical relationships calibrated on measurements (Xu et al., 2016). Note that this
model is considered linear because the degradation due to each cycle is summed over the lifetime. The parameters of the model
are $k_{\delta 1} = 1.4 \times 10^5$, $k_{\delta 2} = -5.01 \times 10^{-1}$, $k_{\delta 3} = -1.23 \times 10^5$, $k_\sigma = 1.04$, $\sigma_{\text{ref}} = 0.5$, $k_T = 6.93 \times 10^{-2}$, $T_{\text{ref}} = 293.15\text{[K]}$ and
$k_t = 4.14 \times 10^{-10}$

$$f^d = \sum_{j=1}^{n_R} \left( (S_{DoD,j} + S_{t_c,j}) \, S_{SoC,j} \times S_{T_c} \right) R_{count,j}$$

$$S_{DoD,j} = (k_{\delta 1} \, R_{DoD,j}{}^{k_{\delta 2}} + k_{\delta 3})^{-1}$$
$$S_{t_c,j} = k_t \, t_{c,j}$$
$$S_{SoC,j}(R_{SoC,j}) = e^{k_\sigma \, (R_{SoC,j} - \sigma_{\text{ref}})}$$
$$S_{T_c} = \begin{cases} e^{k_T \, (T_c - T_{\text{ref}}) \, T_{\text{ref}}/T_c} & \text{if} \quad T_c > T_{\text{ref}} \\ 1 & \text{if} \quad T_c <= T_{\text{ref}} \end{cases}$$

(9)

The non-linear part of the degradation given in Eq. (10) describes the loss of storing capacity (LoC, $L$) using two models:
new battery and used battery after the formation of SEI film. A predefined LoC level is used to determine in which regime the
battery is ($L_1$). $L'$ and $f^{d'}$ are the LoC and linear estimation of LoC when $L$ is equal to $L_1$. Where the parameters of the model
are $\alpha = 0.0575$, $\beta = 121$, and $L_1 = 0.92$.

$$L = \begin{cases} 1 - \alpha \, e^{-\beta f^d} - (1 - \alpha) e^{-f^d} & \text{if} \quad L \leq L_1 \\ 1 - (1 - L') \, e^{-f^d + f^{d'}} & \text{if} \quad L > L_1 \end{cases}$$

(10)

Finally, the time series of the degrading energy capacity of the battery is $B_{E\,deg}(t) = B_{E\,new} \times [1 - L(t)]$. In this article,
the battery degradation model is not coupled to the EMS model, but instead, it uses the resulting state of charge time-series
($SoC(t)$) estimated by the EMS optimization on an operation period (for example, one or two years). The SoC operation
period is repeated to obtain the full lifetime of operation and then used to compute the degradation over the lifetime of the
HPP. Finally, battery replacement occurs when the battery reaches a minimum health level ($1 - L_{min}$). Figure 7 presents a
comparison of the degradation on the battery operating in the same HPP but using different battery fluctuation penalty factors.

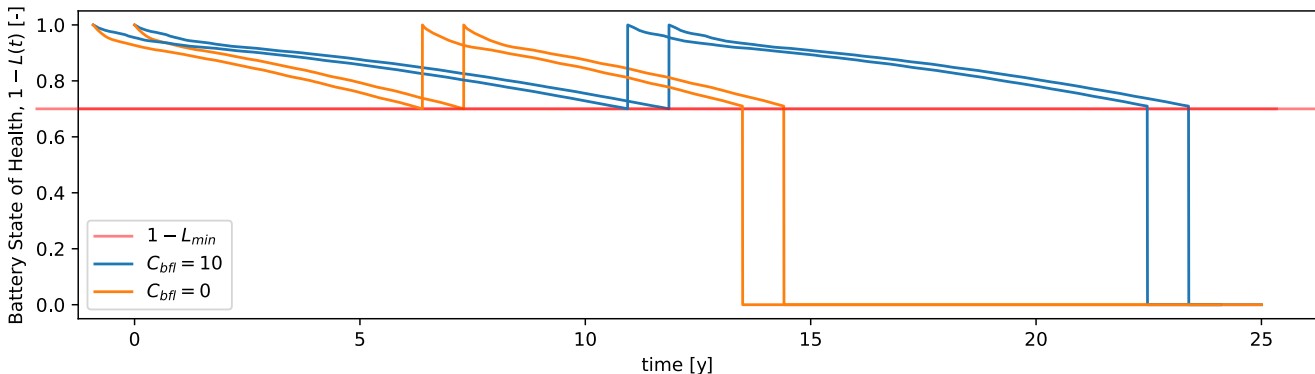

**Figure 7.** Battery degradation comparison in an example HPP for two different battery fluctuation penalty factors $C_{bfl}$.

## 2.10 Long-term operation correction model (EMS Long-term)

A ruled-based EMS is implemented to account for battery, PV, and wind degradation, and forecast errors in estimated wind and solar generation. The correction model consists of the following general principles: (1) try to follow the resulting operation obtained in the EMS described in Section 2.8 ($B(t)$, $E_{SoC}(t)$), (2) update the state of charge to account for the reduction in the available generation in the HPP and the new limits of the degraded battery, (3) recompute the battery power operation and HPP curtailment accounting for the charge and discharge efficiencies.

The implementation consists of computing the reduction in charging power due to the different available generations, as presented in Eq. (11). The SoC ($E_{SoC\,LT}(t)$) is updated, including the constraints of the new energy limits of the degraded battery, Eq. (12). Finally, the battery's power ($B_{LT}(t)$) to supply the SoC, and the curtailment ($P_{\text{curt}\,LT}(t)$) are updated, Eq. (13):

$$B_{LT}^0(t) = \begin{cases} -(W_{\text{deg}}(t) + S_{\text{deg}}(t)) & \text{if} \quad B(t) \leq 0 \quad \text{and} \quad -B(t) > (W_{\text{deg}}(t) + S_{\text{deg}}(t)) \\ B(t) & \text{else} \end{cases} \tag{11}$$

$$E_{SoC\,LT}(t+1) = \begin{cases} E_{SoC\,LT}(t) - \eta_{\text{charge}} B_{LT}^0(t)\,\Delta t & \text{if} \quad B_{LT}^0(t) \leq 0 \\ E_{SoC\,LT}(t) - B_{LT}^0(t)\,\Delta t / \eta_{\text{discharge}} & \text{if} \quad B_{LT}^0(t) > 0 \end{cases}$$
$$E_{SoC\,LT}(t) \geq B_{E\,\text{deg}}(t) \times (1 - B_{E\,\text{depth}})$$
$$E_{SoC\,LT}(t) \leq B_{E\,\text{deg}}(t) \tag{12}$$

$$B_{LT}(t) = \begin{cases} (E_{SoC\,LT}(t) - E_{SoC\,LT}(t+1))/(\eta_{\text{charge}}\,\Delta t) & \text{if} \quad E_{SoC\,LT}(t) - E_{SoC\,LT}(t+1) \le 0 \\ (E_{SoC\,LT}(t) - E_{SoC\,LT}(t+1))/(\Delta t/\eta_{\text{discharge}}) & \text{if} \quad E_{SoC\,LT}(t) - E_{SoC\,LT}(t+1) > 0 \end{cases}$$

$$(13)$$

$$P_{\text{curt}\,LT}(t) = \max(W_{\text{deg}}(t) + S_{\text{deg}}(t) + B_{LT}(t) - G, 0)$$
$$H_{LT}(t) = W_{\text{deg}}(t) + S_{\text{deg}}(t) - P_{\text{curt}\,LT}(t) + B_{LT}(t)$$

## 2.11 Wind plant costs model

A simple WPP cost model consists of estimating the total capital expenditure costs (CAPEX, $C_W$) and operational and maintenance costs (OPEX, $O_W$) as a function of the installed capacity (given as number of turbines times the rated power of the turbines: $W_{\text{MW}} = N_{\text{WT}} P_{\text{rated}}$), the cost of the turbines, their construction and civil infrastructure ($C_{WT} + C_{W\,\text{civil}}$). The OPEX is divided into fixed costs that are scaled with the rated capacity of the plant ($O_{W\,\text{fixed}}$) and variable costs ($O_{W\,\text{var}}$) that scales with the annual energy production of the wind turbines ($AEP_W$) and the ratio between the reference turbine and selected turbine power rating. The wind turbine cost $f_{WT}(D, P_{\text{rated}}, hh)$ (Dykes et al., 2018) depends on the rotor diameter, the WT-rated power, and the tower hub height. This model uses empirical fits to estimate the mass of all WT components and, therefore, for simplicity, is not presented here. The final turbine costs are scaled with respect to the costs of a reference WT ($f_{WT\,\text{ref}}(D_{\text{ref}}, P_{\text{rated\,ref}}, hh_{\text{ref}})$), see Eq. (14).

$$C_W = (f_{WT}/f_{WT\,\text{ref}})(C_{WT} + C_{W\,\text{civil}})\,W_{\text{MW}}$$
$$O_W = W_{\text{MW}} \times O_{W\,\text{fixed}} + AEP_W (P_{\text{rated}}/P_{\text{rated\,ref}})\,O_{W\,\text{var}}$$

$$(14)$$

## 2.12 PV plant costs model

A simple PV plant cost model consists of estimating the total capital expenditure costs (CAPEX, $C_S$) and operational and maintenance costs (OPEX, $O_S$) as a function of the installed capacity ($S_{\text{MW}}$) and solar AC to DC ratio ($r_{\text{AD}}$), see Eq. (15). This model uses the PV costs per megawatt DC ($C_{PV}$), the installation costs per megawatt DC ($C_{S\,\text{install}}$) and fixed operational costs ($O_{S\,\text{fixed}}$), while the inverter costs are provided per megawatt AC ($C_{\text{inv}}$).

$$C_S = (C_{PV} + C_{S\,\text{install}})\,S_{\text{MW}} \times r_{\text{AD}} + C_{\text{inv}} \times S_{\text{MW}}$$
$$O_S = O_{S\,\text{fixed}} \times S_{\text{MW}} \times r_{\text{AD}}$$

$$(15)$$

## 2.13 Battery costs model

The battery plant cost model consists of estimating the total capital expenditure costs (CAPEX, $C_B$) and operational and maintenance costs (OPEX, $O_B$) as a function of the number of batteries required during the plant lifetime ($N_b$, assuming replacement of batteries after degradation) given the new battery energy ($B_E$) and power capacities ($B_P$), see Eq. (16). The

CAPEX model splits the energy capacity costs ($C_{BE}$) and power capacity dependent costs, which include power capacity, installation, and control system costs ($C_{BP} + C_{B\,\text{BOP}} + C_{B\,\text{control}}$). An equivalent number of present batteries ($N_{B\,eq}$) is used to reflect the decrease in costs of battery throughout the lifetime of the battery given a battery price reduction per year ($f_B$) and the time of replacement of the $i_b$) battery in years ($y_b(i_b)$).

$$
\begin{aligned}
C_B &= N_{b\,eq} \times C_{BE} \times B_E + (C_{BP} + C_{B\,\text{BOP}} + C_{B\,\text{control}})\,B_P \\
O_B &= O_{BE} \times B_E \\
N_{B\,eq} &= \sum_{i_b=0}^{N_b-1}(1-f_B)^{y_b(i_b)}
\end{aligned}
\tag{16}
$$

### 2.14 Electrical and shared infrastructure cost model

A simple electrical infrastructure cost model consists of estimating the total capital expenditure costs (CAPEX, $C_E$) as a function of the grid capacity ($G_{MW}$), and the balance of system costs and grid connection costs ($C_{\text{BOS}} + C_{\text{grid}}$) and land costs, see Eq. (17). Note that the HPP land-use area is shared between wind ($A_W$) and solar ($A_S$), given their corresponding installation densities: $\rho_W$ and $\rho_S$.

$$
\begin{aligned}
A_W &= W_{\text{MW}}/\rho_W \\
A_S &= S_{\text{MW}}/\rho_S \\
A_{HPP} &= \max(A_W, A_S) \\
C_E &= (C_{\text{BOS}} + C_{\text{grid}})\,G_{MW} + C_{\text{land}}\,A_{HPP} \\
O_E &= 0
\end{aligned}
\tag{17}
$$

### 2.15 HPP financial model

A simple financial model uses the weighted average cost of capital (WACC) for wind, PV, and battery as a discount rate, see Eq. (18). The WACC after tax ($\text{WACC}_{\text{tx}}$) is the result of weighting sum of the WACCs for wind, PV, battery, and electrical by their corresponding CAPEX, taking the mean WACC for the electrical infrastructure costs shared across all technologies.

$$
\begin{aligned}
C_H &= C_W + C_S + C_B + C_E \\
O_H &= O_W + O_S + O_b + O_E \\
\text{WACC}_m &= (\text{WACC}_W + \text{WACC}_S + \text{WACC}_B)/3 \\
\text{WACC}_{\text{tx}} &= (C_W\,\text{WACC}_W + C_S\,\text{WACC}_S + \\
&\quad C_B\,\text{WACC}_B + C_E\,\text{WACC}_m)/C_H
\end{aligned}
\tag{18}
$$

The financial model then estimates the yearly incomes ($I_y$) and cashflow ($F_y$) as a function of the average revenue over the year, including peak-hour penalties ($R_y = \langle Pr(t)\,H_{LT}(t) - l\rangle_y$), the tax rate ($r_{\text{tax}}$) and $\text{WACC}_{\text{tx}}$. Net present value ($NPV$), the internal rate of return (IRR), and levelized costs of energy ($LCoE$) can then be calculated using the $\text{WACC}_{\text{tx}}$ as the discount rate, see Eq. (19).

$$\begin{aligned}
I_y \quad &= (R_y - O_H)(1 - r_{\text{tax}}) \\
F_y \quad &= \begin{cases} -C_H \text{ for } y = 0 \\[6pt] I_y \text{ for } y > 0 \end{cases} \\[10pt]
NPV \quad &= \sum_y F_y/(1 + \text{WACC}_{\text{tx}})^y \\[8pt]
0 \quad &= \sum_y F_y/(1 + IRR)^y \\[10pt]
C_L \quad &= \sum_y (O_H/(1 + \text{WACC}_{\text{tx}})^y) + C_H \\
AEP_L \quad &= \sum_y (AEP_y/(1 + \text{WACC}_{\text{tx}})^y) \\
LCoE \quad &= C_L/AEP_L
\end{aligned} \tag{19}$$

## 3  Surrogate based optimization

Surrogate-based optimization is used as the outer sizing optimization to reduce the number of full model evaluations during a gradient-based optimization (Jones et al., 1998). In this work, we use the Gaussian process (or Kriging) implementation from the Surrogate Modeling Toolbox (SMT) (Bouhlel et al., 2019). Modern Kriging surrogates with partial least squares-based training (KPLS) are proven to be faster to train and evaluate because of the minimized number of meta-parameters obtained by applying dimensional reduction techniques such principal component analysis to the inputs (Bouhlel et al., 2016b). Furthermore, KPLS can be used to provide near optimal, initial conditions in the training of standard Kriging (KPLSK) (Bouhlel et al., 2016a). KPLSK with squared exponential kernel and linear trend is used as a surrogate model over the design variables.

An updated version of the parallel efficient global optimization (Roux et al., 2020) is proposed to use a two-step approach to (a) explore (find regions with candidates for global optimal) and (b) refine (propose model simulations that help the convergence of EGO optimization on local optima). See Algorithm 1. An initial database of model simulations is generated using Latin hyper-cube sampling (LHS) (McKay et al., 2000; Jin et al., 2003). Then in each optimization iteration, an exploration step identifies regions with candidates for global optimal based on the evaluation of the expected improvement of the surrogate. This is done by parallel execution over $10^4$ random samples (per parallel process) in the design space. Then the top-ranked ($EI_x$) points are clustered using Elkan's K-mean clustering algorithm (Elkan, 2003) and the best performing point per cluster is selected as a candidate ($x_{EI}^+$). A refinement step is performed around the current optimal perturbing of each dimension at a time ($x_{opt}^+$), depending on the iteration convergence the refinement focuses on local perturbations or evaluations of extremes per input dimension. Finally the model is evaluated in parallel ($y^+ \leftarrow \mathcal{M}(x^+)$). The surrogate $\hat{\mathcal{M}}$ is then updated with the updated list of model evaluations ($x^+, y^+$).

**Algorithm 1** Parallel explore and refine EGO algorithm

---

$x = \text{LHS}(n_0)$

$y = \mathcal{M}(x)$                             Initial simulation DB

$x_{opt} = \text{argmin}_x(y)$

**while** $i_{iter} < n_{max\,iter}$ **do**

    $\hat{\mathcal{M}} \leftarrow \text{train}(x, y)$                     Train surrogate model

    $EI_x = \text{EI}(\hat{\mathcal{M}}, x_{opt}, x_x)$         **Explore** the expected improvement

    $x_{EI}^+ \leftarrow \text{get\_candidates}(x_x, EI_x)$       Get optimal candidates based on EI

    **if** $\epsilon \le \epsilon_{tol}$ **then**

        $x_{opt}^+ = \text{perturb\_around\_point}(x_{opt})$     **Refine** around current best

    **else if** $\epsilon > \epsilon_{tol}$ **then**

        $x_{opt}^+ = \text{extremes\_around\_point}(x_{opt})$     **Refine** on single variable extremes

    **end if**

    $x^+ = [x_{EI}^+, x_{opt}^+]$               Concatenate inputs for evaluation

    $y^+ = \mathcal{M}(x^+)$                 Parallel model evaluation

    $x, y \leftarrow [x, x^+], [y, y^+]$          Update model evaluations

    $\epsilon = 1 - y_{opt}/\text{min}(y)$             Update epsilon

    $x_{opt} = \text{argmin}_x(y)$            Update current optimal inputs

    $y_{opt} = \text{min}(y)$                 Update current optimal

**end while**

---

## 4 Study Cases

Three locations in India are selected as study cases, see Fig. 8. These locations are selected because they have a good balance between having good wind resources, good solar resources, or intermediate resources. The wind speed and irradiance statistics are presented in Fig. 9. A summary of costs, assumptions, and specifications used for this analysis are presented in Tables 1 and 2. The costs are taken from DEA Technology Catalogue Danish Energy Agency (2020), while the PV and wind degradation of 0.5 % per year are taken from Theristis et al. (2023) and Hamilton et al. (2020). For each location, the optimization problem is executed based on two different (single) design objectives: $LCoE$ and $NPV/C_H$ in order to illustrate the benefits of HPP design based on revenues. Each optimization is executed with 6 multi-starts in order to ensure global optimality. Finally, we present a sensitivity analysis of the optimization results to varying all battery-related costs by applying a factor.

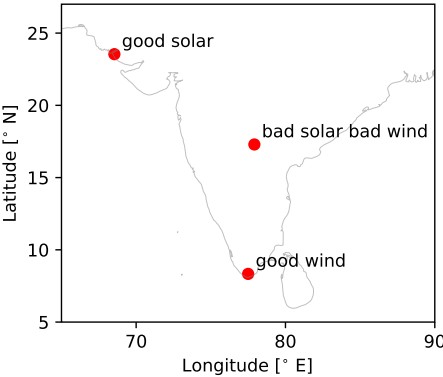

**Figure 8.** Location of the three example sites.

**Figure 9.** Hourly statistics per month for wind speed and direct normal irradiance on the three locations.

**Table 1 (left half)**

| Symbol | Description | Units | Value |
|---|---|---|---|
| | **General** | | |
| $G$ | Grid connection | MW | 300 |
| – | Simulation year | - | 2012 |
| $N_{\text{life}}$ | Lifetime | y | 25 |
| | **WPP** | | |
| $C_{WT}$ | Wind turbine cost | EUR MW$^{-1}$ | 640 000 |
| $C_{W\,\text{civil}}$ | Wind civil works cost | EUR MW$^{-1}$ | 260 000 |
| $O_{W\,\text{fixed}}$ | Wind fixed O&M cost | EUR MW$^{-1}$ y$^{-1}$ | 12 600 |
| $O_{W\,\text{var}}$ | Wind variable O&M cost | EUR MWh$^{-1}$ | 1.35 |
| $D_{\text{ref}}$ | Reference WT diameter | m | 145 |
| $hh_{\text{ref}}$ | Reference WT hub height | m | 100 |
| $P_{\text{rated ref}}$ | Reference WT rated power | MW | 5 |
| $\eta_W$ | WPP efficiency | – | 1 |
| – | Wind degradation curve's year list | y | [0, 25] |
| $dl_W$ | Wind degradation curve | – | [0, 0.125] |
| – | Share between WT deg types | – | 0.5 |
| | **Shared Costs** | | |
| $C_{\text{BOS}}$ | HPP BOS soft cost | EUR MW$^{-1}$ | 119 940 |
| $C_{\text{grid}}$ | HPP grid connection cost | EUR MW$^{-1}$ | 50 000 |
| $C_{\text{land}}$ | Land cost | EUR MW$^{-1}$ | 300 000 |
| | **Finance** | | |
| $\text{WACC}_W$ | Wind WACC | – | 0.052 |
| $\text{WACC}_S$ | Solar WACC | – | 0.048 |
| $\text{WACC}_B$ | Battery WACC | – | 0.08 |
| $r_{\text{tax}}$ | Tax rate | – | 0.22 |
| | **Penalties** | | |
| $P_{r\,\text{peak}} = \text{quant}(P_r, q)$ | Peak hour definition in quantile, $q$ | – | 0.9 |
| $E_{\text{peak req}} = G \times N_h$ | $N_h$ full power hours expected per day at peak price | hours | 2.55 |

**Table 1 (right half)**

| Symbol | Description | Units | Value |
|---|---|---|---|
| | **PV** | | |
| $C_{PV}$ | Solar PV cost | EUR MW$_{\text{DC}}^{-1}$ | 110 000 |
| $C_{S\,\text{install}}$ | Solar hardware installation cost | EUR MW$_{\text{DC}}^{-1}$ | 100 000 |
| $C_{\text{inv ref}}$ | Solar inverter cost | EUR MW$^{-1}$ | 20 000 |
| $r_{AD\,\text{ref}}$ | Ratio AC/DC ref | – | 1.5 |
| $O_{S\,\text{fixed}}$ | Solar fixed O&M cost | EUR MW$_{\text{DC}}^{-1}$ | 4 500 |
| $\rho_S$ | Land use per Solar MW | km$^2$ MW$_{\text{DC}}^{-1}$ | 0.01226 |
| - | Tracking | – | No |
| - | PV degradation curve's year list | y | [0, 25] |
| $dl_S$ | PV degradation curve | – | [0, 0.125] |
| | **BES** | – | |
| $C_{BE}$ | Battery energy cost | EUR MWh$^{-1}$ | 22 500 |
| $C_{BP}$ | Battery power cost | EUR MW$^{-1}$ | 8 000 |
| $C_{BBOP}$ | Battery BOP install. comm. cost | EUR MW$^{-1}$ | 9 000 |
| $C_{B\,\text{control}}$ | Battery control system cost | EUR MW$^{-1}$ | 2 250 |
| $O_{BE}$ | Battery energy O&M cost | EUR MW$^{-1}$ | 0 |
| $B_{E\,\text{depth}}$ | Battery depth of discharge | – | 0.9 |
| $\eta_{\text{charge}}$ | Battery charge efficiency | – | 0.98 |
| $\eta_{\text{discharge}}$ | Battery charge efficiency | – | 0.98 |
| $f_B$ | Battery price reduction per year | – | 0.1 |
| $1 - min(L)$ | Min. level of health | – | 0.7 |
| $N_{B\,\text{max}}$ | Max No. of batteries | – | 5 |
| | **Optimization** | | |
| $N_{\text{procs}}$ | No. of parallel processors | – | 32 |
| $N_{\text{DOE}}$ | No. of initial model evaluations | – | 160 |
| $N_{\text{clusters}}$ | No. of clusters | – | 8 |
| $N_{seed}$ | No. of random starts (seeds) | – | 6 |
| $N_{EI\,\text{pred}}$ | No. of EI predictions per processor | – | $2.5 \times 10^4$ |
| $\epsilon_{tol}$ | Objective function tolerance | – | $1.0 \times 10^{-3}$ |
| $N_{max\,iter}$ | Max. No. of iterations | – | 20 |
| $N_{\text{conv iter}}$ | Min. No. of converged iterations | – | 3 |

**Table 1.** Assumptions for the HPP sizing optimization with two scenarios for battery costs.

| Design variable | Description | Units | Lower Lim. | Upper Lim. | Type |
|---|---|---|---|---|---|
| $h_c$ | clearance | m | 10 | 60 | int |
| $sp$ | specific power | W m$^{-2}$ | 200 | 360 | int |
| $P_{\text{rated}}$ | WT rated power | MW | 1 | 10 | int |
| $N_{\text{WT}}$ | No. WT | – | 0 | 400 | int |
| $\rho_W$ | Wind installation density | MW km$^{-2}$ | 5 | 9 | float |
| $S_{\text{MW}}$ | solar MW | MW | 0 | 400 | int |
| $\theta_{\text{tilt}}$ | PV surface tilt | ° | 0 | 50 | float |
| $\theta_{\text{azim}}$ | PV surface_azimuth | ° | 150 | 210 | float |
| $r_{\text{AD}}$ | DC-AC ratio | – | 1 | 2 | float |
| $B_P$ | Battery power | MW | 0 | 150 | int |
| $B_{E\,h}$ | Battery energy in hours | h | 1 | 10 | int |
| $C_{bfl}$ | cost of battery P fluct. in peak price ratio | – | 0 | 30 | float |

**Table 2.** Design variable in the optimization setup.

 **5  Results**

The detailed results of the hybrid plant sizing optimization based on minimizing $LCoE$ or on maximizing $NPV/C_H$ for the three different locations in India are presented in Table 3. It is observed that batteries are only installed for $NPV/C_H$-based optimal sizing. This is an expected result as batteries add to the costs and do not increase the AEP, besides any curtailment reduction, and therefore do not reduce the $LCoE$. On $NPV/C_H$-optimal plants, the optimizer tries to minimize the penalties

by over-planting the generation and by introducing storage. Over-planting is a concept that has been proposed to increase revenues on WPP when considering losses (Wolter et al., 2020). In general, the $LCoE$-based designs are single-generation technologies because the best performing (lower $LCoE$) energy source is prioritized; a small over-planting is observed to compensate for the degradation over the lifetime. Because the $LCoE$ does not account for the penalties, the $LCoE$-based designs produce negative business cases ($NPV < 0$) for the *good solar* and *bad solar and bad wind* sites. Note that $l_{\text{life}}$ in

Table 3 represents the total penalties summed over the lifetime, and can be twice as large as the total CAPEX on $LCoE$-based designs. $AE_{\text{curt}}$ represents the mean annual energy curtailment and tends to be smaller than the $AEP$ on all sites. The grid utilization factor, defined as the ratio between the mean HPP power and the grid connection ($GUF = \mathbf{E}(H(t))/G$), better captures the capacity factor of an HPP, as it accounts for the energy sold to the grid. It can be seen that the grid utilization factor is larger for $NPV/C_H$-based designs on the solar-driven sites, while it is slightly reduced on the good wind site.

On the *good solar* site, an HPP of PV and storage is obtained for the $NPV/C_H$-based design with significant over-planting, while a single technology PV plant is obtained for the $LCoE$-based design. The PV panel orientation and $r_{\text{AD}}$ are very similar for both cases, but an increase in tilt indicates an effort to increase the generation closer to the morning peak price.

On the *good wind* site, a single wind plant with minimal over-planting is obtained for the $LCoE$-based design, with high-rated power and a tall tower. A hybrid wind, PV, and storage plant with over-planting is selected for $NPV/C_H$-based design.

On this plant, the turbines are smaller with lower towers, and with additional generation produced by PV. The resulting battery power and energy rating are reduced compared to the other sites, which implies that the hybrid generation requires less energy shifting from non-peak to peak hours. On the contrary, this site uses three batteries instead of only two in the other locations. It is interesting to see that both designs on this location have similar final $NPV/C_H$ and $LCoE$ values, highlighting that you can achieve similar objectives with multiple combinations of technologies.

On the *bad solar and bad wind* site, a PV with a storage plant is obtained for the $NPV/C_H$-based design. Note that PV-only plants are, in general, over-planted (320 MW over 300 MW grid), the reason for this is to obtain a better $AEP$ and $GUF$. An example period of operation of the $NPV/C_H$-based HPP for all sites is shown in Fig. 10.

Figure 11 depicts the results of $NPV/C_H$-based optimizations run with varying battery costs. Note that all the battery-related costs are scaled by a unique factor. It can be seen that the cost of batteries has a significant impact on the final HPP

design and performance. The overall business case ($NPV/C_H$) is reduced when the batteries are more expensive for all sites. For both the good solar site and bad wind and bad solar site, the optimal HPP is very similar in terms of wind, solar, and number of batteries. While on the good wind site, batteries are not installed if they are 1.5 more expensive, instead the amount of wind and PV over-planting increases to reduce the penalties and keep a similar business case. Finally, the optimizer decreases the

| Site | | Good solar | | Good wind | | Bad solar bad wind | |
|---|---|---|---|---|---|---|---|
| Design objective | | $LCoE$ | $NPV/C_H$ | $LCoE$ | $NPV/C_H$ | $LCoE$ | $NPV/C_H$ |
| **Design Variables** | **Units** | | | | | | |
| $h_c$ | m | 10 | 10 | 10 | 10 | 10 | 10 |
| $sp$ | $\text{W m}^{-2}$ | 200 | 200 | 360 | 360 | 200 | 200 |
| $P_\text{rated}$ | MW | 1 | 1 | 8 | 4 | 1 | 1 |
| $N_\text{WT}$ | - | 0 | 0 | 38 | 66 | 0 | 0 |
| $\rho_W$ | $\text{MW km}^{-2}$ | 5.0 | 5.0 | 7.8 | 7.4 | 5.0 | 7.5 |
| $S_\text{MW}$ | MW | 322 | 400 | 0 | 54 | 328 | 400 |
| $\theta_\text{tilt}$ | ° | 28.3 | 35.0 | 0.0 | 21.1 | 24.8 | 29.5 |
| $\theta_\text{azim}$ | ° | 210 | 210 | 150 | 210 | 210 | 210 |
| $r_\text{AD}$ | - | 1.5 | 1.6 | 1.0 | 1.7 | 1.7 | 1.9 |
| $B_P$ | MW | 0 | 104 | 0 | 57 | 0 | 150 |
| $b_{Eh}$ | h | 1 | 7 | 4 | 4 | 1 | 7 |
| $C_{bfl}$ | - | 0.0 | 0.0 | 16.0 | 0.7 | 26.7 | 0.0 |
| **Design Summary** | | | | | | | |
| $G$ | MW | 300 | 300 | 300 | 300 | 300 | 300 |
| $W_\text{MW}$ | MW | 0 | 0 | 304 | 264 | 0 | 0 |
| $S_\text{MW}$ | MW | 322 | 400 | 0 | 54 | 328 | 400 |
| $B_P$ | MW | 0 | 104 | 0 | 57 | 0 | 150 |
| $B_E$ | MWh | 0 | 728 | 0 | 228 | 0 | 1050 |
| $N_B$ | - | 0 | 2 | 0 | 3 | 0 | 2 |
| $D$ | m | - | - | 168 | 119 | - | - |
| $hh$ | m | - | - | 94 | 69 | - | - |
| **Outputs** | | | | | | | |
| $NPV/C_H$ | - | -0.264 | 0.747 | 0.996 | 1.042 | -0.548 | 0.537 |
| $NPV$ | MEUR | -42.5 | 178.0 | 304.9 | 304.8 | -96.0 | 151.5 |
| $IRR$ | - | - | 0.128 | 0.145 | 0.151 | - | 0.110 |
| $LCOE$ | $\text{EUR MWh}^{-1}$ | 18.73 | 22.26 | 17.51 | 19.13 | 21.06 | 26.82 |
| $C_H$ | MEUR | 160.9 | 238.3 | 306.2 | 292.6 | 175.1 | 282.3 |
| $O_H$ | MEUR | 2.2 | 2.9 | 5.2 | 6.1 | 2.5 | 3.4 |
| $l_\text{life}$ | MEUR | 372 | 3.8 | 99 | 41 | 417 | 2.9 |
| $AEP$ | GWh | 732 | 927 | 1564 | 1441 | 712 | 918 |
| $AE_\text{curt}$ | GWh | 4.5 | 1.3 | 0.9 | 0.0 | 7.2 | 2.3 |
| $GUF$ | - | 0.28 | 0.35 | 0.60 | 0.55 | 0.27 | 0.35 |
| **Optimization** | | | | | | | |
| Run time | min. | 14 | 19 | 10 | 13 | 9 | 17 |
| No. model eval. | - | 587 | 670 | 485 | 551 | 459 | 641 |

**Table 3.** HPP Sizing optimization results in the example sites with respect $NPV/C_H$ and $LCoE$.

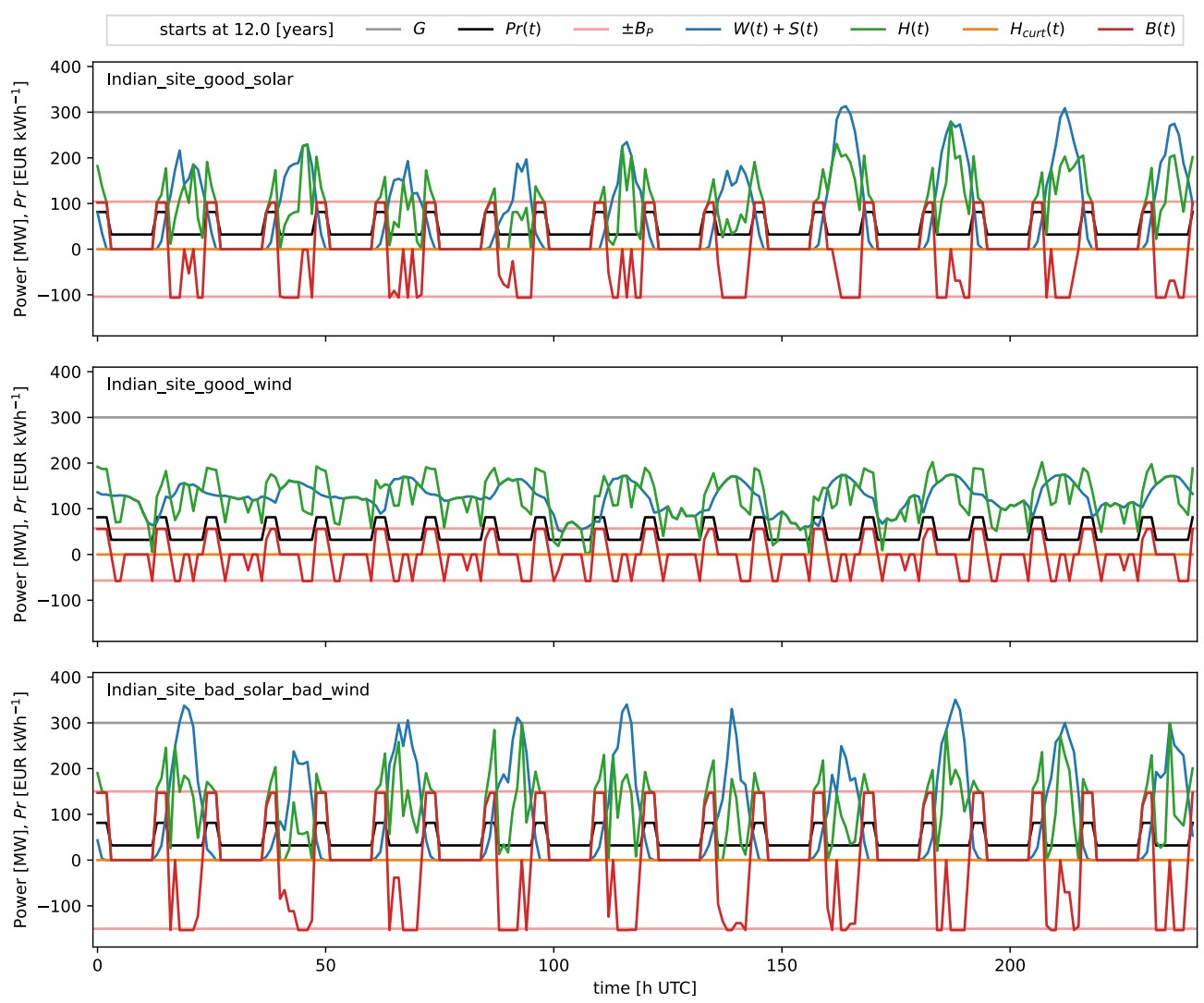

**Figure 10.** Example of 10 days of operation on the 12th year for the $NPV/C_H$-optimized HPPs (top) good solar site (center) good wind site (bottom) bad solar bad wind site.

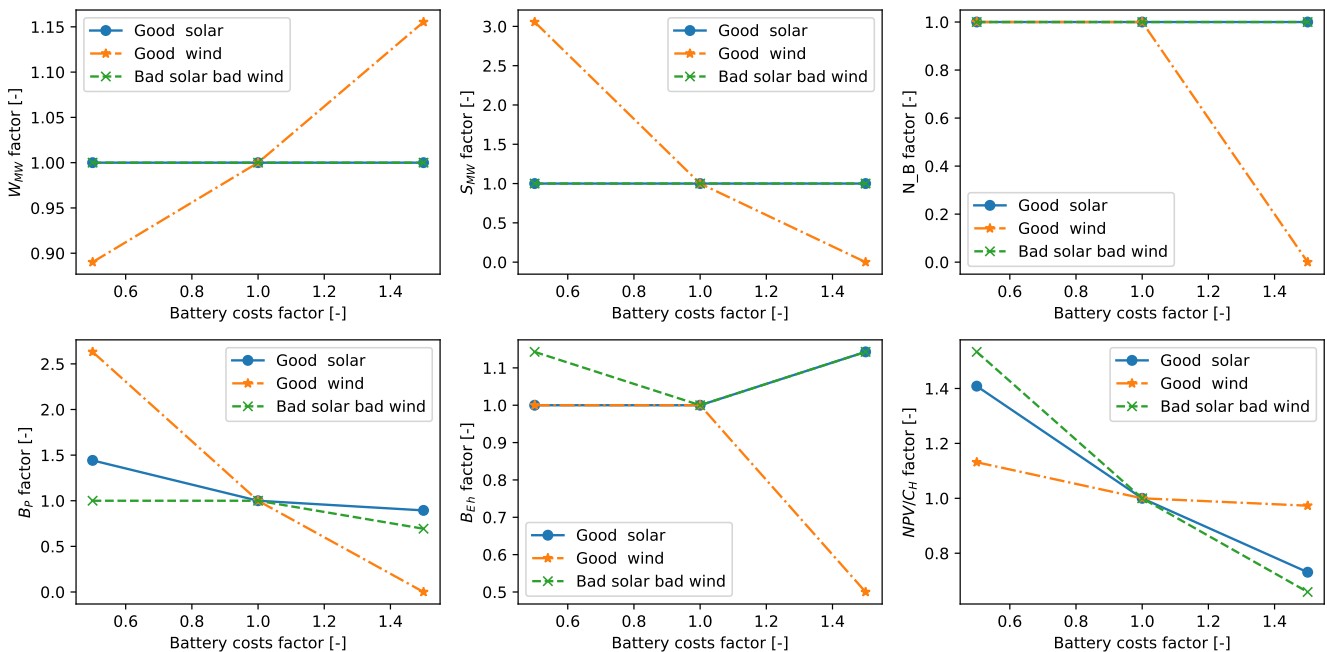

**Figure 11.** Sensitivity of some key outputs for $NPV/C_H$ optimal plants on the three locations when scaling all battery-related costs.

power rating of the batteries when they are more expensive, but a small increase in the energy capacity is seen on the good
solar site and the bad wind site.

## 6 Conclusions and Future Work

Hybrid power plants with storage are obtained across India on $NPV/C_H$-based designs as a consequence of trying to mitigate
the penalties of not reaching the expected energy generation at peak hours. Li-ion batteries are installed on sites that can
not mitigate penalties by over-planting. The results show how changing from $LCoE$ to $NPV/C_H$ driven design allows the
optimizer to over-dimension the generation and include storage to maximize the revenue by balancing the CAPEX, OPEX,
power curtailment, and penalties. Hybrid plants, which include wind, solar, and battery, only occur on sites where the wind and
solar generation complement each other to match the spot price signal (good wind).

Battery degradation plays an important role in HPP sizing as the additional costs of replacing the battery one or two times
will change the financial viability of the project.

The sizing optimization prioritizes cheaper turbines for the $NPV/C_H$-based HPP on the good wind site, by selecting lower
hub height and lower rated power.

The proposed nested optimization approach ensures realistic HPP operation and at the same time allows to have non-linear
sizing optimization. In the proposed framework, both EMS models are necessary since it is not computationally feasible to solve

the internal EMS optimization for varying degradation states for the full lifetime within an outer sizing optimization. Instead, the rule-based long-term EMS is used to account for component degradation in a computationally efficient way. Hybrid power plants should be designed considering a realistic representation of the technologies, including their degradation.

The $IRR$ is not defined when the $NPV$ is negative, but such business cases occur on several HPPs evaluated during a sizing optimization and even on some $LCoE$-optimal HPPs. This illustrates why it is not possible to size HPP sites based on IRR, but instead, we propose the use of $NPV/C_H$ among other modified IRRs.

Future work will look into integrating stochastic optimization with internal operation optimization, to have operation strategies that are robust to the forecast errors. Furthermore, HPP sizing optimization under cost and future spot price uncertainties is planned.

*Code and data availability.* HyDesign is an open source code for the design and control of utility-scale wind-solar-storage based hybrid power plant (HPP). The documentation and example interactive examples are available at(https://topfarm.pages.windenergy.dtu.dk/hydesign/); the input data including weather and price signals for the example Indian sites used in this article are available in the HyDesign repository under examples (https://gitlab.windenergy.dtu.dk/TOPFARM/hydesign).

*Author contributions.* JPM is responsible for the model development, overall implementation, and the article. HH implemented the rule-based correction method. MFM contributed to the implementation of the parallel EGO algorithm. MG contributed to the improvement of the EMS formulation. RZ contributed to the initial implementation of the battery degradation model. KD provided funding and supervision. All authors contributed to the article.

*Competing interests.* The authors declare that there are no competing interests.

*Acknowledgements.* Part of the research was performed in REALISE project funded by EUDP (journal number - 64021-2049) and as part of the Indo-Danish project "HYBRIDize" (https://orbit.dtu.dk/en/projects/optimized-design-and-operation-of-hybrid-power-plant) funded by Danish Innovationsfonden (IFD). KD would also like to acknowledge EUDP IEA Wind Task 50 project for supporting his hours for contributing to the article.

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
