# Peer review of "HyDesign: a tool for sizing optimization of grid-connected hybrid power plants including wind, solar photovoltaic, and Li-ion batteries"

_Wind Energy Science, 2023_

## Referee Comment (RC1)

HyDesign

General:
This study:
1) integrates battery degradation into a hybrid plant sizing optimization problem (through both penalty and integration in the objective function via replacement costs)
2) compares a traditional LCOE objective with an NPV/CAPEX objective to observe the impact on hybrid plant sizing (also includes a penalty for not meeting peak power)
3) applies a novel "efficient global optimization" algorithm with an outer loop of surrogate modeling and inner loop of gradient-based optimization
4) includes optimal EMS integration in hybrid simulation and battery dispatch

Overall, the contributions of this study are noteable and the methods seem thorough. However, the methods are incomplete and, at times, not understandable. The results are barely and poorly presented, and the conclusions section is almost non-existent. I would strongly suggest a major revision of this paper, focusing on:
• clarifying methods and presenting them completely
• presenting results so that major scenarios and findings are compared and impact is detailed, and
• rewriting the conclusion to include major findings and impacts, as well as limitations and future work.
There are also grammatical errors, which should be corrected before the draft is returned to reviewers.

Content:

• Wind and solar both have degradation based on resource, environmental conditions, control, etc. You're right that wind degradation is hard to track given the influence of control, operations, and maintenance on the degradation of the turbine (along with the turbine characteristics, like steel quality). Wind and PV, however, do have open data about efficiency losses, capacity factor decreases, and sometimes degradation rates. You might want to pivot just a little bit from inclusion of PV degradation as a novelty (on line 60) (since the linear degradation is common in solar and hybrid plant modeling) to just dynamic battery degradation with use. I would like to see 1) inclusion of wind degradation, even if it is linear, 2) PV degradation value justification after Line 46 (you state 0.5%, but it isn't cited or justified), and 3) tease out the interaction and further explore the impact of penalizing the ramping of the battery on dynamic battery degradation. The capability that you do have about battery degradation is interesting and worth highlighting more! Can you also speak to the difference in depth of discharge (how you defined the penalty) versus the rate of discharge? Have you limited the rate?
• Could you elaborate on this?: "Note that the EMS optimization is solved using linear programming and therefore does not compute the battery degradation, instead, it assumes new battery and PV panels (without degradation)." Do you mean the EMS assumes a non-degraded battery, even though you are running the battery down to replacement-level degradation? If this is true, it should be noted as a limitation and should detail how it impacts results (potentially assumes more charging capacity than capable, up to 15% capacity (if you replace at 85% max SOC). If it is non-negligible, it should be worth noting why you didn't
• Line 165 talks about this threshold during which the SEI forms, which differentiates two regimes of degradation in LoC. The value of that threshold needs to be given, cited, and justified. So does the linear degradation rate. Any value that is being used should be noted throughout the equations, just as you mention alpha and beta values in line 167.
• Up until page 10, the authors discuss 3 Indian locations to demonstrate their methods. Then, on page 10, 9 new locations are suddenly introduced (in France, the U.K., and Germany) with no justification or background, never to be mentioned again. I would suggest sticking to a smaller set of locations to demonstrate the methods, and to provide justification for why those locations are chosen (like the Indian locations, which compare good solar, good wind, and bad solar/wind conditions). If there is no justification, I suggest leaving the locations out. For the selected locations, it is important to say where those locations are in the respective country, as well. The readers should be able to tell easily where the locations are and why they were chosen.

- On Line 189: there is mention of an EMS comparison, but up to this point, it isn't clear that is one of your objectives (EMS comparison). I see Section 2.7 contains an "EMS Optimization Model" and Section 2.9 contains a "EMS Long-term" model, but I'm unsure of how they fit together. If you are comparing two models, I suggest you combine 2.7 and 2.9 into a single section (move Battery Degradation to 2.7, so that EMS would be Section 2.8), include an intro that describes you are comparing the two EMS methods and why you are doing so, highlighting the major differences. Then, you can have two subsections that describe the models, "EMS optimization model and long-term model". To me, it seems as if section 2.7 already includes battery degradation, so there needs to be some clarification for the reader to follow. Section 2.9 says that the rule-based EMS is implemented to include battery degradation, PV degradation, and forecast errors for wind and solar generation without rerunning the EMS optimizations, which makes me think that the two sections work together to describe a single method, rather than two separate methods for comparison.
- Section 2.9 is difficult to read as written, especially the second paragraph starting at Line 182, and I cannot infer what the authors are trying to say. Figure 7 (Cross-validation errors on rule.) also is difficult to understand. What is the relative error comparing the EMS results against? I understand you are validating the EMS-LT method, but I'm not sure what you are using as your baseline. You should make it clear that you are validating its performance.
- Line 189: By "500 different sizing capacities" do you mean optimization evaluations?
- The scaling relationship you used for the wind turbine cost on Line 201 should be given, even though it is from another paper. All parameter values for the costs models need to be reported.
- When discussing the cost models, the term "user" starts to creep in. It is fine if you are referring to an open-source software model, but up to this point, you've been describing the methods without referencing users and user-defined inputs. For the demonstration of your methods in this paper and the accompanying analysis, you need to provide YOUR inputs. In code documentation, defining user inputs and potential default values is completely appropriate.
- For the cost models, could you elaborate on what goes into the fixed versus variable cost components for CAPEX and OPEX? The reason I ask is that the way you've presented your equations, it hides whether or not you've included replacement costs. Because degradation is a central part of your paper's stated novelty, it is worthwhile to include how replacement was integrated. I see you have called it out with the N_b variable in Section 2.12 for Battery costs, so that is great. How you calculate N_b needs to be detailed (read: at what point is replacement triggered? What is that threshold?).
- Might want to reiterate on line 235 that you are comparing two different objectives, your novel use of NPV/CAPEX to the traditional LCoE to see differences in performance regarding resulting technology capacities.
- What does "outer" mean in Line 242? When you define the "efficient global optimization" algorithm in the abstract, you say, "This new algorithm is a surrogate-based optimization method." There is no mention of a nested approach. I think it would be good to clearly outline to the reader earlier in the paper when it is mentioned (both abstract and intro) that it is a nested optimization approach that uses an outer surrogate-based optimization loop, followed by a gradient-based optimization inner loop.
- Table 1 finally defines parameters used in the methods, but is incomplete (doesn't define all variables), uses parameter names that are inconsistent with how they were defined, and doesn't use the same styling (for instance, using the Euro symbol or spelling out Euro). It also introduces a comparison not ever mentioned in the rest of the paper, and that is "expensive batteries" versus "cheap batteries." This comparison of scenarios NEEDS to be introduced and justified earlier when the scope of the paper is being presented in the introduction. Why is this comparison included? Now we are back to the 3 Indian sites as well, and we've ignored the 9 other sites? How does the cost of more expensive batteries compare to the savings in replacement costs due to degradation?
- Line 267: "Over-planting the generation or by introducing storage" maybe should be "over-building generation and storage capacity"? "Over-planting" is not a common term and comes off as odd. Used again on L269, L270, etc. Please check throughout paper.
- Line 269: What is meant by "business case is negative"? Can you elaborate, using specific results, what is makes it a negative business case? Used again on L274.

- Line 271: What is meant by, "The final size is a combination of reductions of land costs and wake losses, as it can be seen in the selection of larger spacing ($\rho_W$) for the LCoE-based design."? Final size of which case and location?
- Line 277: Grid utilization factor needs to be defined so that we understand the result and its impact.
- Tables and figures should appear in the order they are mentioned, which would put Figure 8 below the tables that are mentioned before it in text.
- In general, the results and conclusions are lacking and require significant revisions. There are so many aspects of this paper that are completely ignored in the results section. The link between the methods and the impact of the results are not fully explored in presented results nor in text. For each "novelty" claimed in the abstract and each "scenario" defined in the methods section, the results need to support each of those aspects of the paper. What was the impact of including degradation in battery modeling? What was the impact of including more or less expensive batteries? What was the impact of building a site for NPV/CAPEX rather than LCOE (this is barely touched on)? What do all these findings mean for developers and the wider research community? How should we be designing plants, based on your findings? Each finding should have supporting text describing what is driving the finding. For example, if cheap batteries are more advantageous, why? Does the CAPEX of the batteries outweigh the replacement costs for cheaper batteries? At what price does that change?
- The conclusion section seems incomplete. It needs to outline the contributions of the paper, both from a methodological and from a results standpoint, describe limitations, and propose future work to overcome those limitation and continue to advance the field.

Grammar: Note that many of these corrections include an example but should be applied throughout the paper
- Consistent italisizing/capitalization of names (ex: Line 75 has pywake and Line 81 has *pyWake*)
- Equations should have all variables defined, and then the figures should consistently use those variables (rather than units). For instance, Equation 1 has some variables defined, but others not (like N_WT), and then in Figure 3, MW/km^2 is used rather than the variable defined rho_W. Ideally, similar subscripts would be awesome, but I know Matplotlib is limited.
- Consistently capitalize "Figure" and "Equation" rather than "figure" and "equation" (ex: Line 71 has "figure 1" but Line 90 has "Figure 4".
- Consistently use undercase in captions. For instance, "Figure 2. Generic Wind Turbine surrogate" should be "Generic wind turbine surrogate"
- Define acronyms before use (examples include CAPEX, which is used on Line 47 and defined on line 196 and WPP, which is first used in Figure 1 and Line 85 and is defined on Line 111, SEI on Line 165, WT on line 201)
- L 127 Spot to spot
- L 129 Has to have
- Spacing around equations is a bit odd (particularly, there seems to be extra space above, maybe?)
- Line 72 potential spelling error: constrain > constraint
- Line 44 potential grammar error: "while *the* wind turbine" rather than "while wind turbine"
- Line 121, need a comma between the two subscripts "S" and "inverter" so that it is "C_S, inverter
- Line 119, missing an article and comma at "grid-connected": "Sandia performance model for *a* grid-connected, PV-inverter model"
- Figure 6 caption: "two…factor*s"* rather than "factor"
- You use "solar PV" and "PV", so I suggest you pick one for consistency
- Line 132: Once you define an abbreviation, use it consistently (ex: EMS is defined early in the doc, but then spelled out throughout). Definition of abbreviations only need to happen once (ex: OPEX, CAPEX, EMS, and others are defined multiple times). Also, "The energy management system optimization model consists in" should be "The energy management system optimization model consists of"
- Line 156: "a rainflow counting" should be "Rainflow Counting…is implemented" or "A Rainflow Counting algorithm/method…is implemented"
- Line 177: "A ruled based EMS is implemented" should be "A rule-based EMS is implemented"
- Line 180: "*the* reduction"…missing article 'the'

- Line 191: "Spot electricity price" > "spot market price", "single year operations" > "single-year operations"
- Throughout paper: "Consist in" should be "consists of"
- Line 199: "fix costs" > "fixed costs"
- Line 208: ", solar AC to DC ratio" > "and solar AC to DC ratio"
- Line 209: "inverter costs is" > "inverter costs are"
- Line 247: "near optimal initial conditions" > "near-optimal, initial conditions"…this grammar issue is throughout and should be thoroughly checked.
- Line 261: "A summary of assumptions costs and general specifications of HPP" > "A summary of assumptions, costs, and general specifications used for this analysis"
- Line 267: Wind is capitalized when "w" should be undercased
- Line 274: missing comma after PV. You can decide to use the Oxford comma or not, but just do it consistently.
- Line 278: "This is in general an expected result" > clauses need commas for separtation > "This is, in general, and expected result"
- Figure 8 caption: "10 days *of* operation" and "NPV-optimized"

---

## Referee Comment (RC2)

**HyDesign: a tool for sizing optimization for grid-connected hybrid power plants including wind, solar photovoltaic, and Li-ion batteries**

This paper presents a modelling framework, coupled to an optimizer, that can be used to optimize various essential components and the control strategy of a co-located hybrid power plant. It includes performance and cost models for various disciplines along with degradation effects of PV panels and Li-ion batteries. The use of NPV normalized with investment as an objective function, instead of LCoE, makes the results relevant to the wind-based hybrid community.

Overall, the study uses a comprehensive modelling approach with case studies to demonstrate the capabilities of the framework. However, the presentation of results and the translation of results to high level insights is missing. The paper needs major revisions w.r.t. the following aspects:

- Paper structure: The methodology section discusses the 'modelling framework'. The general problem formulation (which is currently section 3) is essentially a part of the methodology. It might be useful to have a separate section called ' case study description' that shows 'weather' and 'electricity price' specifically for the locations in India.
- Results section: A detailed discussion w.r.t. the results is missing.
- Conclusions: The conclusions are a short summary of the results, and do not provide any additional higher level insights. For instance, a discussion about how various scenarios/objectives/model parameters drive the HPP design would be useful.

Specific comments:

**Abstract:** Some information about key findings from the application of the tool should also be mentioned.

**1.Introduction**

Is the objective to have a modelling framework that includes effects that are mostly missing in other modelling frameworks commonly used in literature? Maybe it's better to explicitly state the research objective post line 60.

**2.Methodology**

- Before diving into the models, having the general problem formulation where the objective function and the design variables (Current Section 3) are defined would be more helpful.

- The interaction between components, like turbines shadowing the panels, for instance, are not included. That's fine but can be added to the simplifications in line 74.
- Section 2.2, a brief explanation of the behaviour observed in Figure 3 and 4 is needed. Is min_WT_spacing a consequence of the generated layout or is it a constraint? In Figure 4, the wake losses for the lowest specific power turbine are the highest in the partial load region. Is that because it has a larger diameter and hence, lower normalized spacing for the same absolute spacing?
- Section 2.5, the plant capacity in equation 4, $S_{MW}$, is not introduced in the text.
- Section 2.5, some additional information on the reason for degradation along with the rationale behind using 0.5% must be given.
- Section 2.6, as mentioned, the elements specific to the case study can be a separate section. The source for the PPA prices must be mentioned. Values of 200-300 Euros/kWh (in Figure 5) look extremely high. Are the units of $Pr$ correct?
- Section 2.7, the grid capacity ($G$) used in equation 5 is not introduced before.
- Sections 2.7-2.9 are a bit difficult to follow. Section 2.7 discusses a revenue-maximizing control strategy where a perfect forecast is assumed. The factor $C_{bfl}$ also prevents the ramping of batteries and in a way, accounts for battery degradation in the form of an economic penalty. Section 2.8 then discusses a detailed degradation model but its implementation is not clear. Does it use the SoC from the idealized EMS operation to calculate the health followed by the replacement of batteries (and hence added CAPEX) every time the battery reaches 70% state of health?
- The purpose of Section 2.9 is not clear. It is called 'long-term' operation but it looks like it mainly deals with imperfect forecast of wind and solar. Figure 7 and the paragraph before is not clear and needs to be better explained. Is it an update of idealized EMS or is it a new model? If $r_{EMS}$ is the revenue using perfect forecast, is the purpose of this Figure 7 to show the difference in revenue introduced because of imperfect forecast?
- For Section 2.10, it might be necessary to provide more cost details. If a reference turbine is used, mention its characteristics and the source. Since these costs have a major impact on the economic metric, it's useful to see how the turbine cost factor ($f_{WT}$) changes/scales with $D$, $P$, and $hh$. I'm also not sure why $f_{WT}$ is multiplied with both $C_{WT}$ and $C_{Wcivil}$. For the OPEX, scaling the variable part with AEP is a bit unfair because the O&M costs might depend more on the number of turbines, spare part cost of the RNA, etc. But with this model, a farm with a lower specific turbine will have a much higher OPEX even if other factors (number of turbines, farm rated power) are the same. It may be fine for this work but its better to be aware of how the model choice drives OPEX and hence, the conclusions.
- The "user" of the tool may not always be the same. It depends on the stakeholder that uses the tool and inputs provided will change depending on who is using the tool.
- Section 2.13, the variable (ele_cost) in the section heading might be a typo.
- In Section 2.14, if NPV already discounts the cashflows with the $WACC_{tx}$, I'm not sure why the net yearly revenue ($I_y$) is multiplied again by ($1-WACC_{tx}$).

**3.HPP Sizing optimization**

- Turbine's specific power (sp) is commonly expressed in [W/m$^2$]. [MW/m$^2$] might be fine but line 236 uses [m$^2$/MW].
- The equality constraints shown in equation 17 are not really constraints but intermediate variables that are derived from the design variables.

**5.Results**

- As stated before, a separate case study section is needed that specifically discusses 'India-specific' elements like the prices, resources, grid capacity, etc. Also, the reader should know that the resources, battery prices, etc. are going to be varied. At the moment, it suddenly shows up in the Results.
- Line 273: "*On the bad solar and bad wind site, a hybrid wind, PV and storage plant is selected for the NPV/C$_H$-based design with a marginally positive business case. This illustrates it is not possible to size HPP sites based on IRR, there are several configurations that will produce negative business cases and therefore have undefined IRR.*" I'm not sure how the conclusion about IRR follows from the previous line. IRR, as a metric, is quite similar to NPV/C$_H$ and they should have similar results. A design resulting in a positive NPV/C$_H$ will also result in an IRR higher than the WACC. You could use MIRR instead of IRR to avoid certain issues with IRR but both MIRR and NPV/C$_H$ should have a similar behaviour for negative business cases, as long as MIRR values are feasible.
- Line 279. It makes sense that an LCoE-optimized design does not result in a mix of technologies. But in case of NPV/C$_H$, there was an incentive to have a high GUF in the form of pricing. In case of LCoE, does it make sense to have something equivalent in the form of a capacity factor constraint?

**6.Conclusions**

This section needs to be completely revisited.
- Discuss how the different economic metrics drive HPP design and make recommendations that can be useful for the community.
- How do different resources drive the design? The study currently just mentions good and bad resource but are there more mechanisms? Like the anti-correlation between wind and solar.
- The effect of including battery degradation and different battery costs needs to be better explained.
- How do pricing mechanisms drive the HPP. Is the HPP a result of the manner in which the price incentives are defined in this study?

**Grammar and styling:**

Some sentences use 'a HPP' (line 65) while HPP is defined as 'Hybrid power plants' in the first sentence. I suggest you drop the plural and use it consistently throughout the paper.

- Line 24: 'Sizing of HPP plant….'.  HPP already includes plant.
- Line 24: MDAO is Multi-disciplinary Design Analysis and Optimization
- Line 28: 'hybrid pant sizing as an MDAO problem including..'
- Line 47: Introduces CAPEX but is not defined.
- Line 69: 'In the sizing optimization, several…'
- Line 105: 'ERA5 (0.1 degrees instead of 0.25 degrees in latitude and longitude resolution), and it shows a better validation metric for individual plant modelling.' Not sure if I understand the second part of the sentence though.
- Introduce $\theta_{zenith}$, used in equation 2, in the text.
- Line 109: 'The DHI is estimated using the solar zenith angle ($\theta_{zenith}$), as shown in equation 2.'
- Line 150: Remove spacing in the brackets. ($E_{SOC}(t)$) and ($B_E(t)$).
- The subscripts in the variables that are descriptive shouldn't be italicized. For example, $P_{curt}$ (t) in line 141, $Pr_{max}$ in line 146, $\eta_{charge}$, $\eta_{discharge}$ in line 151. This is currently used throughout the paper and needs to be corrected.
- Consistently use 'Figure', 'Equation', 'Table' throughout the paper.
- Line 275-277: 'This illustrates why 275 it is not possible to size HPP sites based on IRR, there are several configurations that will produce negative business cases and therefore have undefined IRR. Note that PV-only plants are in general over-planted (320 MW over 300 MW grid), the reason for this is to obtain a better annual energy production (AEP) and grid utilization factor (GUF).' Reframe. Either start a new sentence or use a conjunction.
- Line 282: An NPV/$C_H$ based design is possible, but not an NPV/$C_H$ based site. Rephrase.

---

## Author Comment (AC1)

**Reply to the reviewer's comments of article (reviewer 1)**
Hydesign: a tool for sizing optimization for grid-connected hybrid power plants including wind, solar photovoltaic, and Li-ion batteries

The authors would like thank the reviewer for providing detailed and thorough comments that have made the article better. We have addressed all your comments. In this document you will find the reviewer comments in red and a summary of the modifications done including line number references of where to find them on the new document.

**General**:

This study:

- integrates battery degradation into a hybrid plant sizing optimization problem (through both penalty and integration in the objective function via replacement costs)
- compares a traditional LCOE objective with an NPV/CAPEX objective to observe the impact on hybrid plant sizing (also includes a penalty for not meeting peak power)
- applies a novel "efficient global optimization" algorithm with an outer loop of surrogate modeling and inner loop of gradient-based optimization
- includes optimal EMS integration in hybrid simulation and battery dispatch

Overall, the contributions of this study are noteable and the methods seem thorough. However, the methods are incomplete and, at times, not understandable. The results are barely and poorly presented, and the conclusions section is almost non-existent. I would strongly suggest a major revision of this paper, focusing on:

- clarifying methods and presenting them completely
- presenting results so that major scenarios and findings are compared and impact is detailed, and
- rewriting the conclusion to include major findings and impacts, as well as limitations and future work.

There are also grammatical errors, which should be corrected before the draft is returned to reviewers.

**Reply to general comments**:

- The *Methods* section has been revised and rewritten to make sure that the methods are understandable, reproducible and complete.
- A new Section 5 *Study Cases* presents the examples sites, their specific input data, including an overview of their weather. See Figures 8 and 9. The *HPP sizing optimization* section that describes the outer problem formulation is now part of the *Methods*, section 2.1.
- The *Results* section has been rewritten to highlight the results. Key results are presented in Table 3 for the main costs scenario. While a sensitivity analysis of the sizing optimization to changes in costs of batteries is presented in Figure 11.
- The *Conclusions and Future Work* section has been rewritten to highlight major findings,

and to include limitations and future work.

**Content:**

- Wind and solar both have degradation based on resource, environmental conditions, control, etc. You're right that wind degradation is hard to track given the influence of control, operations, and maintenance on the degradation of the turbine (along with the turbine characteristics, like steel quality). Wind and PV, however, do have open data about efficiency losses, capacity factor decreases, and sometimes degradation rates. You might want to pivot just a little bit from inclusion of PV degradation as a novelty (on line 60) (since the linear degradation is common in solar and hybrid plant modeling) to just dynamic battery degradation with use. I would like to see 1) inclusion of wind degradation, even if it is linear, 2) PV degradation value justification after Line 46 (you state 0.5%, but it isn't cited or justified), and 3) tease out the interaction and further explore the impact of penalizing the ramping of the battery on dynamic battery degradation. The capability that you do have about battery degradation is interesting and worth highlighting more! Can you also speak to the difference in depth of discharge (how you defined the penalty) versus the rate of discharge? Have you limited the rate?

- A linear degradation (measured as loss in capacity factor) on the wind turbine has been implemented (See Equation 5 and Figure 5). This method ensures a linear degradation on the capacity factor over the age of the wind farm by mixing two mechanisms of degradation: (a) a shift in the power curve towards higher wind speeds represents blade degradation and increasing friction losses. (b) a loss factor applied to the power time series represent increase in availability losses.

- Citations to the PV degradation rate of 0.5%/year are given in Use Cases section. Sizing optimization of hybrid power plant usually does not consider the effect that degradation on the generation have on the long-term operation of the hybrid power plant with storage. We have re-phrase the novelties.

- We have clarified the explanation of the battery degradation in the article as well as the explanation of the impact of penalizing the ramping on the battery degradation.

- Depth of discharge is a constraint forced to the battery (as specified by the manufacturer) that consist of ensuring a minimum level of energy in the battery.

- The rate or ramping in the power provided (or received) by the battery is used to implemented a penalty to the objective function of the (internal) operation optimization. the penalty ensures that excessive changes in battery power (such as intermittent charge-discharge) do not occur. This penalty has as a consequence an extension of the battery life. We have clarified that this penalty is an input to the operation (internal) optimization, while it is a design variable for the sizing (external) optimization (Equation 1).

- The battery degradation model has been updated to remove the need to have discrete levels of health (See Figure 7).

- Could you elaborate on this?: "Note that the EMS optimization is solved using linear programming and therefore does not compute the battery degradation, instead, it assumes new battery and PV panels (without degradation)." Do you mean the EMS assumes a non-degraded battery, even though you are running the battery down to replacement-level degradation? If this is true, it should be noted as a limitation and should detail how it impacts results (potentially assumes more charging capacity than capable, up to 15% capacity (if you replace at 85% max SOC). If it is non-negligible, it should be worth noting why you didn't.

- The internal operation optimization or Energy management system (EMS) is implemented using linear programming (LP). The battery degradation is highly non-linear because it requires the computation of load cycles using the Rainflow counting algorithm, and it has non-linear dependency to its load cycles.

- The EMS then does assume new batteries. To account for the degradation of PV, wind and battery in the operation we implemented a long-term operation model that has as an objective to account for the degradation in generation as well as the loss of storing capacity due to battery degradation. The long-term operation model is applied as a correction to the "planned" idealized operation of the new battery, as it tries to follow the planned SoC but has less generation available and less storage capacity to do so. The description of this two components has been expanded in the article (Sections 2.8, 2.9 and 2.10).

- Line 165 talks about this threshold during which the SEI forms, which differentiates two regimes of degradation in LoC. The value of that threshold needs to be given, cited, and justified. So does the linear degradation rate. Any value that is being used should be noted throughout the equations, just as you mention alpha and beta values in line 167.

- The empirical values for the battery degradation models are provided in Section 2.9.

**Section 2.9 on the submitted article:**

- Up until page 10, the authors discuss 3 Indian locations to demonstrate their methods. Then, on page 10, 9 new locations are suddenly introduced (in France, the U.K., and Germany) with no justification or background, never to be mentioned again. I would suggest sticking to a smaller set of locations to demonstrate the methods, and to provide justification for why those locations are chosen (like the Indian locations, which compare good solar, good wind, and bad solar/wind conditions). If there is no justification, I suggest leaving the locations out. For the selected locations, it is important to say where those locations are in the respective country, as well. The readers should be able to tell easily where the locations are and why they were chosen.

- On Line 189: there is mention of an EMS comparison, but up to this point, it isn't clear that is one of your objectives (EMS comparison). I see Section 2.7 contains an "EMS Optimization Model" and Section 2.9 contains a "EMS Long-term" model, but I'm unsure of how they fit together. If you are comparing two models, I suggest you combine 2.7 and 2.9 into a single section (move Battery Degradation to 2.7, so that EMS would be Section 2.8), include an intro that describes you are comparing the two EMS methods and why you are doing so, highlighting the major differences. Then, you can have two subsections that describe the models, "EMS optimization model and long-term model". To me, it seems as if section 2.7 already includes battery degradation, so there needs to be some clarification for the reader to follow. Section 2.9 says that the rule- based EMS is implemented to include battery degradation, PV degradation, and forecast errors for wind and solar generation without rerunning the EMS optimizations, which makes me think that the two sections work together to describe a single method, rather than two separate methods for comparison.

- Section 2.9 is difficult to read as written, especially the second paragraph starting at Line 182, and I cannot infer what the authors are trying to say. Figure 7 (Cross-validation errors on rule.) also is difficult to understand. What is the relative error comparing the EMS results against? I understand you are validating the EMS-LT method, but I'm not sure what you are using as your baseline. You should make it clear that you are validating its performance.

- Line 189: By "500 different sizing capacities" do you mean optimization evaluations?

The section (Section 2.9 of the original submitted article) that compares the two EMS methodologies has been removed as it miss-leads the reader. Note that:

- Both EMS models are necessary since it is not computationally feasible to solve the EMS optimization for varying degradation states for the full lifetime within an outer sizing optimization. Instead the rule-based long-term EMS is used to account for the degradation and forecast error in a computationally efficient way.

- The comparison between solving an EMS problem for degraded components and the proposed correction to the idealized EMS was done for different degradation conditions on HPPs on nine locations (some in Europe). This locations were not discussed in the article and therefore misleads the reader as the article only presents and discusses results in the specified three Indian locations.

- The two EMS models solve different problems. The EMS optimization solves the operation of HPP with a battery for a representative period (one up to five years). This idealized operation is assumed to be repeated for the lifetime of the project and it is used to estimate the degradation of the components (the battery degradation depends on how you operate it, while the PV and wind degradation are driven only by the age of the plant).

- As mentioned before, the long-term EMS operation model receives the idealized or "planned" operation and tries to follow it as much as possible. The actual operation is different because there is degradation on the generation and battery. The long-term operation then provides the income of the hybrid power plant for all the life-time.

- The reason to include the battery degradation model in between the two EMS sections is because the degradation is a required input to the long term operation evaluation.

- The old section 2.9 was intended to demonstrate that the long-term operation component achieves similar revenues than solving a new EMS operation optimization given the degraded components. We have removed the section because it makes it hard to understand the need for both models.

- 500 different sizing capacities are used to represent different levels of degradation, for example a new battery with 5 hours of energy storage is equivalent to a battery of 10 hours with a degradation of 50%.

- The scaling relationship you used for the wind turbine cost on Line 201 should be given, even though it is from another paper. All parameter values for the costs models need to be reported.

- When discussing the cost models, the term "user" starts to creep in. It is fine if you are referring to an open-source software model, but up to this point, you've been describing the methods without referencing users and user-defined inputs. For the demonstration of your methods in this paper and the accompanying analysis, you need to provide YOUR inputs. In code documentation, defining user inputs and potential default values is completely appropriate.

- For the cost models, could you elaborate on what goes into the fixed versus variable cost components for CAPEX and OPEX? The reason I ask is that the way you've presented your equations, it hides whether or not you've included replacement costs. Because degradation is a central part of your paper's stated novelty, it is worthwhile to include how replacement was integrated. I see you have called it out with the $N_b$ variable in Section 2.12 for Battery costs, so that is great. How you calculate $N_b$ needs to be detailed (read: at what point is replacement triggered? What is that threshold?).

- The wind turbine cost scaling relationship has been reported in more details. But because the WT cost model uses empirical fits to estimate the mass of all WT components, the model is not presented in detail in the article.

- We have remove references to users in the article. But in practice there are several assumptions that are provided as inputs to the model. We have summarized the assumptions including cost assumptions in Tables 1 and 2.

- CAPEX includes replacement costs of batteries, but because battery replacements occur during the lifetime of the project we have included an assumption of decreasing costs of battery that depreciates the future costs to bring it as a present value at the beginning of the project. We have clarify the mechanism for battery replacement in the section of battery degradation. See Equation 15.

- Might want to reiterate on line 235 that you are comparing two different objectives, your novel use of NPV/CAPEX to the traditional LCoE to see differences in performance regarding resulting technology capacities.

- What does "outer" mean in Line 242? When you define the "efficient global optimization" algorithm in the abstract, you say, "This new algorithm is a surrogate-based optimization method." There is no mention of a nested approach. I think it would be good to clearly outline to the reader earlier in the paper when it is mentioned (both abstract and intro) that it is a nested optimization approach that uses an outer surrogate- based optimization loop, followed by a gradient-based optimization inner loop.

- We have included both clarifications: (a) We describe the comparison of the optimization results using two different objective functions. (b) We make it clearer in the abstract and outline that the methodology is a nested optimization approach.

- Table 1 finally defines parameters used in the methods, but is incomplete (doesn't define all variables), uses parameter names that are inconsistent with how they were defined, and doesn't use the same styling (for instance, using the Euro symbol or spelling out Euro). It also introduces a comparison not ever mentioned in the rest of the paper, and that is "expensive batteries" versus "cheap batteries." This comparison of scenarios NEEDS to be introduced and justified earlier when the scope of the paper is being presented in the introduction. Why is this comparison included? Now we are back to the 3 Indian sites as well, and we've ignored the 9 other sites? How does the cost of more expensive batteries compare to the savings in replacement costs due to degradation?

- We have re-edited Table 1 to make it consistent with the article in variable name and units notation.

- The purpose of the comparison is to demonstrate how the optimal sizing of a HPP changes with cheaper costs of battery. We have change the article to use an unique scenario (Table 1) and its results are given in Section 5 and Table 3. A sensitivity to changes in costs of batteries is now presented as a plot (Figure 11).

- Component degradation (battery, wind and pv) reduce the revenues, while battery replacement adds to the CAPEX of the battery. Having more expensive batteries forces the optimization to reduce the power and energy ratings of the battery or to use the battery more conservatively to make it last longer.

- Line 267: "Over-planting the generation or by introducing storage" maybe should be "over-building generation and storage capacity"? "Over-planting" is not a common term

and comes off as odd. Used again on L269, L270, etc. Please check throughout paper.

- A reference to a publication that describes over-planting as an approach to maximize revenues on wind power plants has been included ([Wolter et al., 2020]).

- Line 269: What is meant by "business case is negative"? Can you elaborate, using specific results, what is makes it a negative business case? Used again on L274.

- Line 271: What is meant by, "The final size is a combination of reductions of land costs and wake losses, as it can be seen in the selection of larger spacing ($\rho_W$) for the LCoE-based design."? Final size of which case and location?

- Line 277: Grid utilization factor needs to be defined so that we understand the result and its impact.

- Tables and figures should appear in the order they are mentioned, which would put Figure 8 below the tables that are mentioned before it in text.

- We have added a clarification of what negative business case means ($NPV < 0$), Section 5.

- We have rewritten Section 5.

- We have added a definition of GUF Section 5.

- Tables and figures are now displayed in the referred order.

- In general, the results and conclusions are lacking and require significant revisions. There are so many aspects of this paper that are completely ignored in the results section. The link between the methods and the impact of the results are not fully explored in presented results nor in text. For each "novelty" claimed in the abstract and each "scenario" defined in the methods section, the results need to support each of those aspects of the paper.

- What was the impact of including degradation in battery modeling?

- What was the impact of including more or less expensive batteries?

- What was the impact of building a site for NPV/CAPEX rather than LCOE (this is barely touched on)?

- What do all these findings mean for developers and the wider research community? How should we be designing plants, based on your findings? Each finding should have supporting text describing what is driving the finding.

- For example, if cheap batteries are more advantageous, why? Does the CAPEX of the batteries outweigh the replacement costs for cheaper batteries? At what price does that change?

- The conclusion section seems incomplete. It needs to outline the contributions of the paper, both from a methodological and from a results standpoint, describe limitations, and propose future work to overcome those limitation and continue to advance the field.

- The Results and Conclusions sections have been re-written and extended.

- Battery degradation is required to determine the number of batteries needed, and in order to capture the trade-off between revenues and battery life reduction.

- Having more expensive batteries reduces the business cases, and can even produce hybrid plants without storage.

Grammar: Note that many of these corrections include an example but should be applied throughout the paper

- Consistent italisizing/capitalization of names (ex: Line 75 has pywake and Line 81 has pyWake)

- Equations should have all variables defined, and then the figures should consistently use those variables (rather than units). For instance, Equation 1 has some variables defined, but others not (like $N_{WT}$), and then in Figure 3, $MW/km^2$ is used rather than the variable defined $\rho_W$. Ideally, similar subscripts would be awesome, but I know Matplotlib is limited.

- Consistently capitalize "Figure" and "Equation" rather than "figure" and "equation" (ex: Line 71 has "figure 1" but Line 90 has "Figure 4".

- Consistently use undercase in captions. For instance, "Figure 2. Generic Wind Turbine surrogate" should be "Generic wind turbine surrogate"

- Define acronyms before use (examples include CAPEX, which is used on Line 47 and defined on line 196 and WPP, which is first used in Figure 1 and Line 85 and is defined on Line 111, SEI on Line 165, WT on line 201)

- Consistency in italisizing/capitalization of names has been checked.

- All variable names are defined and are used consistently in the text, equations, figures and tables.

- Capitalization of references to equations, figures and tables has been checked.

- Consistent use of under-case in captions.

- Definition of acronyms has been checked.

- L 127 Spot to spot. Corrected

- L 129 Has to have. Corrected

- Spacing around equations is a bit odd (particularly, there seems to be extra space above, maybe?) Checked

- Line 72 potential spelling error: constrain ¿ constraint Sentence has been re-written

- Line 44 potential grammar error: "while the wind turbine" rather than "while wind turbine" Corrected

- Line 121, need a comma between the two subscripts "S" and "inverter" so that it is "$C_S$, inverter. Corrected

- Line 119, missing an article and comma at "grid-connected": "Sandia performance model for a grid-connected,PV-inverter model". Corrected

- Figure 6 caption: "two...factors" rather than "factor". Corrected

- You use "solar PV" and "PV", so I suggest you pick one for consistency. Corrected

- Line 132: Once you define an abbreviation, use it consistently (ex: EMS is defined early in the doc, but then spelled out throughout). Definition of abbreviations only need to happen once (ex: OPEX, CAPEX, EMS, and others are defined multiple times). Corrected

- Also, "The energy management system optimization model consists in" should be "The energy management system optimization model consists of". Corrected

- Line 156: "a rainflow counting" should be "Rainflow Counting...is implemented" or "A Rainflow Counting algorithm/method...is implemented". Corrected

- Line 177: "A ruled based EMS is implemented" should be "A rule-based EMS is implemented". Corrected

- Line 180: "the reduction"...missing article 'the'. Corrected

- Line 191: "Spot electricity price" ¿ "spot market price", "single year operations" ¿ "single-year operations". Does not apply

- Throughout paper: "Consist in" should be "consists of". Corrected

- Line 199: "fix costs" ¿ "fixed costs". Corrected

- Line 208: ", solar AC to DC ratio" ¿ "and solar AC to DC ratio". Corrected

- Line 209: "inverter costs is" ¿ "inverter costs are". Corrected

- Line 247: "near optimal initial conditions" ¿ "near-optimal, initial conditions"...this grammar issue is throughout and should be thoroughly checked. Corrected

- Line 261: "A summary of assumptions costs and general specifications of HPP" ¿ "A summary of assumptions, costs, and general specifications used for this analysis" Corrected

- Line 267: Wind is capitalized when "w" should be undercased. Corrected

- Line 274: missing comma after PV. You can decide to use the Oxford comma or not, but just do it consistently. Checked on all the document

- Line 278: "This is in general an expected result" ¿ clauses need commas for separtation ¿ "This is, in general, and expected result". Corrected

- Figure 8 caption: "10 days of operation" and "NPV-optimized"

---

## Author Comment (AC2)

**Reply to the reviewer's comments of article (reviewer 2)**
**Hydesign: a tool for sizing optimization for grid-connected hybrid power plants including wind, solar photovoltaic, and Li-ion batteries**

The authors would like thank the reviewer for providing detailed and thorough comments that have made the article better. We have addressed all your comments. In this document you will find the reviewer comments in red and a summary of the modifications done including line number references of where to find them on the new document.

**General**:

This paper presents a modelling framework, coupled to an optimizer, that can be used to optimize various essential components and the control strategy of a co-located hybrid power plant. It includes performance and cost models for various disciplines along with degradation effects of PV panels and Li-ion batteries. The use of NPV normalized with investment as an objective function, instead of LCoE, makes the results relevant to the wind-based hybrid community. Overall, the study uses a comprehensive modelling approach with case studies to demonstrate the capabilities of the framework. However, the presentation of results and the translation of results to high level insights is missing. The paper needs major revisions w.r.t. the following aspects:

- Paper structure: The methodology section discusses the 'modelling framework'. The general problem formulation (which is currently section 3) is essentially a part of the methodology. It might be useful to have a separate section called ' case study description' that shows 'weather' and 'electricity price' specifically for the locations in India.

- Results section: A detailed discussion w.r.t. the results is missing.

- Conclusions: The conclusions are a short summary of the results, and do not provide any additional higher level insights. For instance, a discussion about how various scenarios/objectives/model parameters drive the HPP design would be useful.

**Reply to general comments**:

- The *Methods* section has been revised and rewritten to make sure that the methods are understandable, reproducible and complete.

- A new Section 5 *Study Cases* presents the examples sites, their specific input data, including an overview of their weather. See Figures 8 and 9.

- The *Results* section has been rewritten to highlight the results. Key results are presented in Table 3 for the main costs scenario. While a sensitivity analysis of the sizing optimization to changes in costs of batteries is presented in Figure 11.

- The *Conclusions and Future Work* section has been rewritten to highlight major findings, and to include limitations and future work.

**Specific comments**:

- Abstract: Some information about key findings from the application of the tool should also be mentioned.

- The abstract now includes an overview of the results and conclusions.
**1.Introduction**

- Is the objective to have a modelling framework that includes effects that are mostly missing in other modelling frameworks commonly used in literature? Maybe it's better to explicitly state the research objective post line 60.

- The research objective is to build a framework for optimization of hybrid power plants that can be extended to include: sizing and physical design. This sentence has been added in the Introduction.

  **2.Methodology**

- Before diving into the models, having the general problem formulation where the objective function and the design variables (Current Section 3) are defined would be more helpful.

- The *HPP sizing optimization* section that describes the outer problem formulation is now part of the *Methods*, section 2.1.

- The interaction between components, like turbines shadowing the panels, for instance, are not included. That's fine but can be added to the simplifications in line 74.

- A sentence has been added.

- Section 2.2, a brief explanation of the behaviour observed in Figure 3 and 4 is needed. Is min_WT_spacing a consequence of the generated layout or is it a constraint?

- A sentence has been added to Section 2.3.

- In Figure 4, the wake losses for the lowest specific power turbine are the highest in the partial load region. Is that because it has a larger diameter and hence, lower normalized spacing for the same absolute spacing?

- This has been added. That is correct. For two plants with turbines with the same rated power, number of turbines and installation density (and therefore same plant area), the one that has lower specific power will have larger rotor diameter and therefore less WT spacing and higher wake losses. Note also that CT is the same for at low wind speeds for all turbines.

- Section 2.5, the plant capacity in equation 4, SMW, is not introduced in the text.

- A sentence has been added to Section 2.6.

- Section 2.5, some additional information on the reason for degradation along with the rationale behind using 0.5% must be given.

- References have been added to the assumptions in Section 4.

- Section 2.6, as mentioned, the elements specific to the case study can be a separate section. The source for the PPA prices must be mentioned. Values of 200-300 Euros/kWh (in Figure 5) look extremely high. Are the units of Pr correct?

- Section 4 provides a description to the use cases. Note that the price in Figure 6 is the black line and take values of 32.1 or 82.4 Euros/kWh.

- Section 2.7, the grid capacity (G) used in equation 5 is not introduced before.

- Definition added in Section 2.8.

- Sections 2.7-2.9 are a bit difficult to follow. Section 2.7 discusses a revenue-maximizing control strategy where a perfect forecast is assumed. The factor $C_{bfl}$ also prevents the ramping of batteries and in a way, accounts for battery degradation in the form of an economic penalty.

- Section 2.8 then discusses a detailed degradation model but its implementation is not clear. Does it use the SoC from the idealized EMS operation to calculate the health followed by the replacement of batteries (and hence added CAPEX) every time the battery reaches 70% state of health?

- The purpose of Section 2.9 is not clear. It is called 'long-term' operation but it looks like it mainly deals with imperfect forecast of wind and solar. Figure 7 and the paragraph before is not clear and needs to be better explained. Is it an update of idealized EMS or is it a new model? If rEMS is the revenue using perfect forecast, is the purpose of this Figure 7 to show the difference in revenue introduced because of imperfect forecast?

- Sections 2.8, 2.9 and 2.10 have been re-written.

- The internal EMS operation optimization model does not compute the degradation, and it assumes non degraded wind, PV and batteries over a shorter period (in this case one year). The resulting idealized operation is repeated over the lifetime.

- The battery degradation model uses the SoC history from the idealized to compute the battery degradation.

- Both EMS models are necessary since it is not computationally feasible to solve the EMS optimization for varying degradation states for the full lifetime within an outer sizing optimization. Instead the rule-based long-term EMS is used to account for the degradation and forecast error in a computationally efficient way.

- The long-term EMS operation model receives the idealized or "planned" operation and tries to follow it as much as possible. The actual operation is different because there is degradation on the generation and battery. The long-term operation then provides the income of the hybrid power plant for all the life-time.

- The paragraph in section (Section 2.9 of the original submitted article) that compares the two EMS methodologies has been removed as it miss-leads the reader. The comparison between solving an EMS problem for degraded components and the proposed correction to the idealized EMS was done for different degradation conditions on HPPs on nine locations (some in Europe). This locations were not discussed in the article and therefore misleads the reader as the article only presents and discusses results in the specified three Indian locations.

- For Section 2.10, it might be necessary to provide more cost details. If a reference turbine is used, mention its characteristics and the source. Since these costs have a major impact on the economic metric, it's useful to see how the turbine cost factor (fWT) changes/scales with D, P, and hh. I'm also not sure why fWT is multiplied with both CWT and CWcivil. For the OPEX, scaling the variable part with AEP is a bit unfair because the O&M costs might depend more on the number of turbines, spare part cost of the RNA, etc. But with this model, a farm with a lower specific turbine will have a much higher OPEX even if other factors (number of turbines, farm rated power) are the same. It may be fine for this work but its better to be aware of how the model choice drives OPEX and hence, the conclusions.

- Costs and technical characteristics of all technologies are taken from the Danish Energy Agency Catalogue. A reference to it has been added to Section 4, and Table 1 has been reformatted.

- The "user" of the tool may not always be the same. It depends on the stakeholder that uses the tool and inputs provided will change depending on who is using the tool.

- All references to users have been removed.

- Section 2.13, the variable (ele_cost) in the section heading might be a typo.

- This has been removed.

- In Section 2.14, if NPV already discounts the cashflows with the WACCtx, I'm not sure why the net yearly revenue (Iy) is multiplied again by (1-WACCtx).

- This has been removed.

**3.HPP Sizing optimization**

- Turbine's specific power (sp) is commonly expressed in [W/m2]. [MW/m2] might be fine but line 236 uses [m2/MW].

- This typo has been removed.

- The equality constraints shown in equation 17 are not really constraints but intermediate variables that are derived from the design variables.

- The definition of the intermediary variables has been moved to their corresponding components.

**5.Results**

- As stated before, a separate case study section is needed that specifically discusses 'India-specific' elements like the prices, resources, grid capacity, etc. Also, the reader should know that the resources, battery prices, etc. are going to be varied. At the moment, it suddenly shows up in the Results.

- The specific inputs and assumptions for the use cases are described in Section 4.

- Line 273: "On the bad solar and bad wind site, a hybrid wind, PV and storage plant is selected for the NPV/CH-based design with a marginally positive business case. This illustrates it is not possible to size HPP sites based on IRR, there are several configurations that will produce negative business cases and therefore have undefined IRR." I'm not sure how the conclusion about IRR follows from the previous line. IRR, as a metric, is quite similar to NPV/CH and they should have similar results. A design resulting in a positive NPV/CH will also result in an IRR higher than the WACC. You could use MIRR instead of IRR to avoid certain issues with IRR but both MIRR and NPV/CH should have a similar behaviour for negative business cases, as long as MIRR values are feasible.

- It is true that if you defined a modified IRR to remove the problems when the $NPV < 0$, then it should behave similarly to the $NPV/C_H$. The statement explain why we are not using IRR.

- Line 279. It makes sense that an LCoE-optimized design does not result in a mix of technologies. But in case of NPV/CH, there was an incentive to have a high GUF in the form of pricing. In case of LCoE, does it make sense to have something equivalent in the form of a capacity factor constraint?

- This is an interesting idea that we will explore in future work.

**6.Conclusions**
This section needs to be completely revisited.

- The Conclusion section has been rewritten.

- Discuss how the different economic metrics drive HPP design and make recommendations that can be useful for the community.

- This has been added in Section 6.

- How do different resources drive the design? The study currently just mentions good and bad resource but are there more mechanisms? Like the anti-correlation between wind and solar.

- This has been added to the discussion of the results, in particular for the good wind site when the battery costs are high, in Section 5.

- The effect of including battery degradation and different battery costs needs to be better explained.

- This has been added in Section 5 and 6.

- How do pricing mechanisms drive the HPP. Is the HPP a result of the manner in which

the price incentives are defined in this study?

- This is true, the penalty to provide energy at peak hours is one of the main drivers for the hybridization. This has been discussed in Section 5 and 6.

**Grammar and styling:**

- Some sentences use 'a HPP' (line 65) while HPP is defined as 'Hybrid power plants' in the first sentence. I suggest you drop the plural and use it consistently throughout the paper.

- This has been corrected in the Introduction.

- Line 24: 'Sizing of HPP plant....'. HPP already includes plant.

- This has been corrected.

- Line 24: MDAO is Multi-disciplinary Design Analysis and Optimization

- This has been added.

- Line 28: 'hybrid pant sizing as an MDAO problem including..'

- The sentence has been rewritten.

- Line 47: Introduces CAPEX but is not defined.

- This has been corrected.

- Line 69: 'In the sizing optimization, several...'

- The sentence has been rewritten.

- Line 105: 'ERA5 (0.1 degrees instead of 0.25 degrees in latitude and longitude resolution), and it shows a better validation metric for individual plant modelling.' Not sure if I understand the second part of the sentence though.

- This has been rewritten.

- Introduce $\theta_{zenith}$ , used in equation 2, in the text. Line 109: 'The DHI is estimated using the solar zenith angle ($\theta_{zenith}$), as shown in equation 2.'

- This has been added.

- Line 150: Remove spacing in the brackets. (ESOC(t)) and (BE(t)).

- This has been corrected.

- The subscripts in the variables that are descriptive shouldn't be italicized. For example, Pcurt(t) in line 141, Prmax in line 146, $\eta$charge, $\eta$discharge in line 151. This is currently used throughout the paper and needs to be corrected.

- This has been corrected.

- Consistently use 'Figure', 'Equation', 'Table' throughout the paper.

- This has been corrected.

- Line 275-277: 'This illustrates why 275 it is not possible to size HPP sites based on IRR, there are several configurations that will produce negative business cases and therefore have undefined IRR. Note that PV-only plants are in general over-planted (320 MW over 300 MW grid), the reason for this is to obtain a better annual energy production (AEP) and grid utilization factor (GUF).' Reframe. Either start a new sentence or use a conjunction.

- This sentence has been rewritten and moved to conclusions.

- Line 282: An NPV/CH based design is possible, but not an NPV/CH based site. Rephrase.

- This has been corrected.

---

## Author Response (AR2)

February 13, 2024

Reply to the reviewer's comments version two of the article:
**Hydesign: a tool for sizing optimization of grid-connected hybrid power plants including wind, solar photovoltaic, and Li-ion batteries**

The authors would like thank the reviewer for providing detailed and thorough comments that have made the article better. We have addressed the following problems commented after the first review:

1. Untis are handled following the requirements of WES: Units are written in full when used without a number, and written as negative exponents when used with numbers, and in tables and figures.

2. Spelling and typos have been fixed.

3. Color-blind friendly color-schemes have been used.